# NRF2 activators and the inhibitor of nuclear export, selinexor, restrict coronaviruses by targeting a network involving ACE2, TMPRSS2, and XPO1 through an NRF2-independent mechanism

Fakhar H. Waqas[1,2], Leandro Silva da Costa [3], Francisco J. Zapatero-Belinchón [4,10],
Madalina E. Carter-Timofte[3], Lisa Lasswitz[4], Demi van der Horst[3,11], Rebecca Möller[4,5], Julia Dahlmann[6,7],
Ruth Olmer[6,7], Robert Geffers[8], Gisa Gerold[4,12], David Olagnier [3] & Frank Pessler [1,2,9] ✉

Nuclear factor erythroid 2–related factor 2 (NRF2) plays important roles in antiviral host cell defenses. We assessed the potential of the NRF2 activators 4-octyl itaconate (4OI), bardoxolone (BARD), and sulforaphane (SFN), and the exportin-1 (XPO1) blocker selinexor (SEL) to inhibit highly pathogenic (SARS-CoV-2) and seasonal (hCoV-229E) coronaviruses in cellular models. We find that NRF2 knock-out enhances infection by both viruses, but that the compounds restrict these viruses in a largely NRF2-independent manner. 4OI and SEL are most effective against SARS-CoV-2 when added to media before infection, and they reduce cell entry of SARS-CoV-1 and -2 spike protein VSV pseudotypes >10-fold. Strikingly, the compounds downregulate *ACE2, TMPRSS2*, and *XPO1* mRNA and protein, whereby 4OI diminishes STAT3 phosphorylation and represses the *XPO1* gene promoter. 4OI dramatically reduces ACE2 half-life, which requires ubiquitin E3 ligases NEDD4L and MDM2, but is mediated by the lysosomal pathway. XPO1 knock-down reduces CoV-229E replication and reveals that efficacy of the compounds against CoV-229E depends on XPO1 expression in the order SEL > 4OI > SFN > BARD, suggesting that especially BARD restricts hCoV-229E via another, unknown, target. Taken together, these results suggest that "NRF2 activators" can restrict human coronaviruses by targeting an NRF2-independent network involving ACE2, TMPRSS2, and XPO1.

Highly pathogenic human coronaviruses (SARS-CoV-1, SARS-CoV-2, MERS) can cause severe infection characterized by hyperinflammation and end-organ damage such as acute respiratory distress syndrome and multi-organ failure, particularly in individuals with poor pre-existing immunity or medical risk factors such as obesity, diabetes, or cardiovascular disease[1] so called seasonal, low pathogenic coronaviruses (e.g., hCoV-229E, -OC43, -NL63, -HKU1) usually cause mild upper respiratory infections, but they can, nonetheless, cause severe disease in immunocompromised individuals. Direct-acting antivirals such as remdesivir and nirmatrelvir/ritonavir have been approved for the treatment of SARS-CoV-2 infection[2] (update accessed Dec. 4, 2024). However, considering

the strong contribution of dysfunctional inflammatory responses to organ pathology, adjunct therapy with corticosteroids, IL-6 blockade (tocilizumab), and/or the JAK1/2 inhibitor baricitinib is recommended by the World Health Organization[2] (update accessed 4 Dec 2024). In addition to direct-acting antivirals and immunomodulatory adjunct treatments, host-directed antivirals would constitute a third line in the armamentarium against coronaviruses. Host-directed antivirals are compounds that stimulate endogenous antiviral mechanisms or interfere with host cell functions that are required for viral infectivity. The KEAP1/NRF2 signaling pathway activates cytoprotective and anti-inflammatory responses in host cells, but also has antiviral functions in itself[3–5].

---

Activators of this pathway, therefore, constitute promising host-directed interventions for acute viral infections. Indeed, antiviral effects of NRF2 agonists have been demonstrated against a broad spectrum of human pathogenic viruses, including SARS-CoV-2[6], HSV[6], and influenza A[5,7,8]. However, in most cases, it is not known whether the antiviral effects truly depend on the induction of NRF2 signaling or whether other targets contribute. Indeed, in our recent study of NRF2 activators as anti-influenza agents we found that the antiviral effect of the "classic" NRF2 activators bardoxolone-methyl (BARD), sulforaphane (SFN), and 4-octyl itaconate (4OI) did not require functional NRF2-signaling, but was mediated largely by inhibiting nuclear export of viral ribonucleoprotein by the nuclear export factor exportin-1 (XPO1, also known as chromosome region maintenance 1, CRM1)[5]. We also proposed a structural explanation for why the compounds recognize such diverse targets: they can bind covalently to the functionally critical cysteine 528 of XPO1 because it is displayed in a structural context very similar to cysteine 151 in their canonical binding site on KEAP1, the major cellular inhibitor of NRF2[5]. Interestingly, pharmacologic XPO1 inhibition also reduces infectivity of SARS-CoV-2 in vitro and in vivo, even though coronaviruses replicate in the cytoplasm, suggesting that XPO1 mediates shuttling of viral proteins or nuclear export of host cell factors required for optimal infectivity[9,10].

In the current study, we assessed the relative contributions of activating NRF2 signaling and inhibiting XPO1-mediated nuclear export to anti-coronavirus activity. Specifically, we evaluated the ability of these three NRF2 activators and the selective inhibitor of nuclear export selinexor (SEL) to reduce infectivity of SARS-CoV-2 and the low-pathogenic human coronavirus (HCoV)-229E and tested whether the observed antiviral effects indeed depended on the presumed targets, i.e., NRF2 and XPO1. We find that all compounds exert substantial anti-coronavirus effects, but that these are independent of NRF2 signaling and are in part due to targeting XPO1 and, in the case of SARS-CoV-1 and -2, their cellular receptors ACE2 and TMPRSS2.

## Results

### Inhibition of SARS-CoV-2 is NRF2 independent

We first compared the ability of the NRF2 activators (BARD, SFN, 4OI) to inhibit SARS-CoV-2 infection in Calu-3 cells 48 h post-infection (p.i.). LDH release assay demonstrated that the compounds were non-toxic to Calu-3 cells after 48 h of incubation (Supplementary Fig. S1). All three compounds significantly reduced viral copy number in cell supernatants and lysates (Fig. 1A, B). SARS-CoV-2 infection strongly induced expression of interferon (IFN) regulated genes *IFIT1* and *CXCL10* (Fig. 1C, D). Consistent with

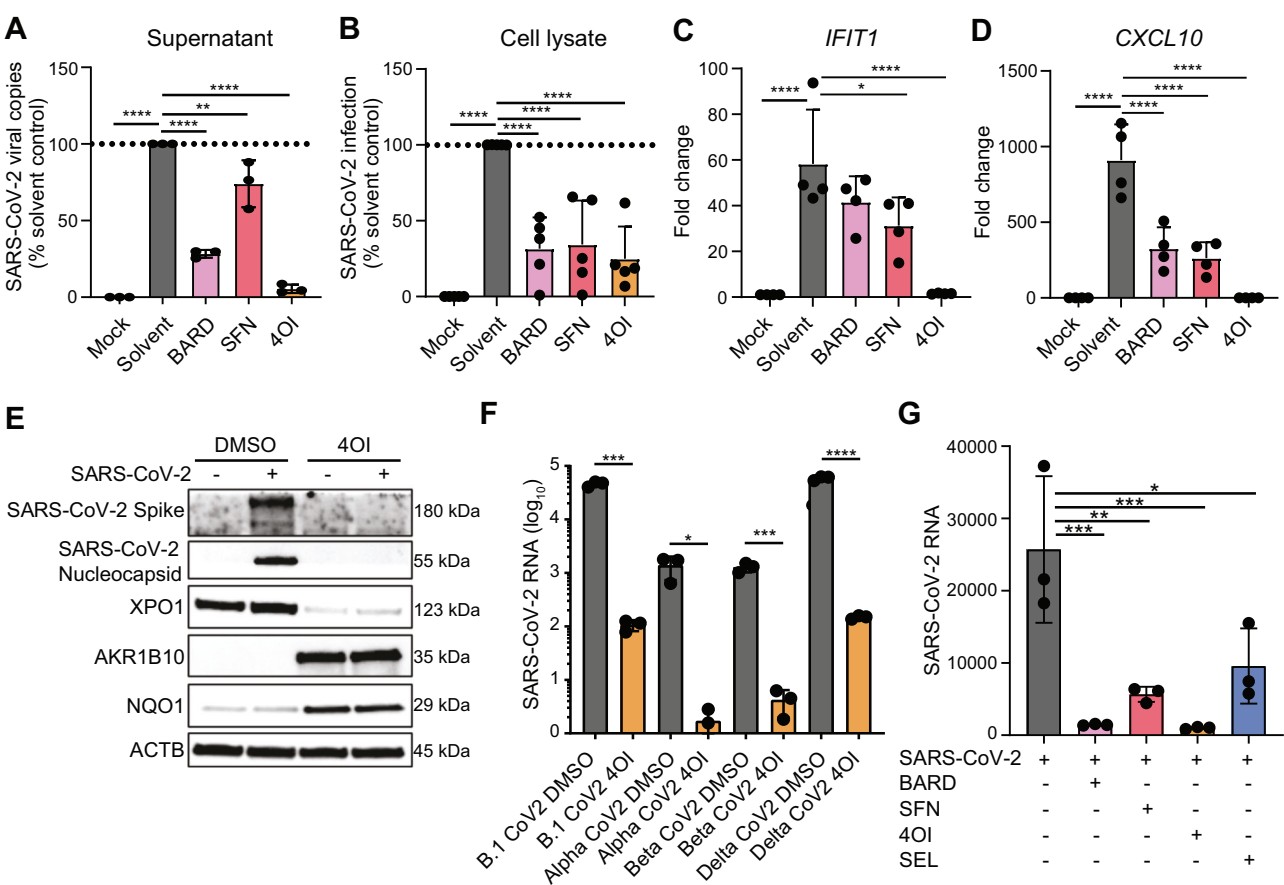

**Fig. 1 | NRF2 activators (4OI, BARD, SFN) and SEL reduce SARS-CoV-2 infectivity in Calu-3 cells.** Experiments for panels A-D were performed in the Gerold lab (Hannover, Germany) and for panels E-G in the Olagnier lab (Aarhus, Denmark). **A–D** Calu-3 cells were pretreated with the compounds (BARD, 0.1 μM; SFN, 10 μM; 4OI, 100 μM) for 24 h, inoculated with SARS-CoV-2/München-1.2/2020/984,p3 (MOI = 0.005) in the presence of the compounds for 4 h, followed by removing the viral inoculum and adding fresh medium containing the respective compounds and controls. Measurements were made 48 h p.i. **A, B** Viral genome copies in supernatants (**A**) and cell lysates (**B**) (RT-qPCR). **C, D**. *IFIT1* and *CXCL10* mRNA in cell lysates (RT-qPCR). Calu-3 cells were pretreated with the indicated compounds for 48 h (**E, F**) or 24 h (**G**), infected with SARS-CoV-2 Wuhan-like early

European B.1 lineage (FR-4286) (MOI = 0.01) for 1 h, followed by removal of the inoculum and incubation in fresh medium containing the compounds. Target gene expression and protein levels were measured 24 h p.i. In **F** infections were additionally performed with the SARS-CoV-2 variants indicated on the x-axis (MOI = 0.01). **E** Reduction of SARS-CoV-2 spike and nucleocapsid proteins, and XPO1 protein expression, but increase in AKR1B10 and NQO1 levels by 4OI (125 μM) (immunoblot with β-actin as internal reference). **F** Marked reduction of viral genomic RNA of diverse SARS-CoV-2 variants of concern by 4OI (125 μM). **G** Marked reduction of viral genomic RNA by the compounds (RT-qPCR normalized against *TBP* mRNA). n = 3, means ± SEM. One-way ANOVA with Tukey's post-hoc test. * ≤0.05, ** ≤0.01, *** ≤0.001, **** ≤0.0001.

their antiviral effects, all compounds significantly reduced expression of both mRNAs, whereby 4OI led to the greatest reduction, possibly due to a combination of stronger antiviral capacity and its well-documented anti-IFN effect[7,11–13]. The antiviral effect of 4OI was also verified by immunoblot. 4OI treatment essentially prevented accumulation of SARS-CoV-2 spike and nucleocapsid protein and led to higher levels of the NRF2-inducible proteins AKRB1 and NQO1 independent of SARS-CoV-2 infection, underscoring its ability to activate NRF2 signaling in this infection model (Fig. 1E). Since 4OI blocks XPO1 and XPO1 inhibition can lead to reduced XPO1 protein levels[14], we also measured XPO1 expression by immunoblot. Indeed, XPO1 levels were markedly lower under 4OI treatment (Fig. 1E), which agreed well with the findings by Ribo-Molina et al. that 4OI treatment reduced XPO1 levels in MDCK and Jurkat cells[15]. 4OI efficiently restricted infectivity of SARS-CoV-2 isolates belonging to the α, β, and δ clades in Calu-3 cells independent of their ability to replicate in untreated cells (Fig. 1F), demonstrating that the antiviral effect of at least 4OI applied to SARS-CoV-2 isolates from different clades. Considering the observed downregulation of XPO1 by 4OI, we then compared the XPO1 inhibitor SEL against the NRF2 activators. SEL did reduce SARS-CoV-2 RNA levels more than 50%, but it was clearly less effective than 4OI and BARD (Fig. 1G). These results suggest that another target, besides XPO1, contributes to the anti-SARS-CoV-2 effects of at least 4OI and BARD.

### The antiviral effect is strongest during viral entry and is largely NRF2 independent

To test at which step the compounds interfered with SARS-CoV-2 infection, we performed an order-of-addition experiment (Fig. 2A). 4OI and SEL were most effective when added to culture media before infection, which was seen both with NP levels in cells (Fig. 2B, C) and viral titers in supernatants (Fig. 2D–G). Comparing infectivity in 786-O renal cell carcinoma cells expressing either WT NRF2 or harboring a stable CRISPR-Cas NRF2 KO[16], revealed significantly higher levels of NP in untreated KO cells, underscoring the importance of endogenous NRF2 in restricting this virus. Of note, 4OI reduced NP expression in both WT and NRF2 KO cells, and, again, its antiviral effect was strongest during pre-infection (Fig. 2H). These results suggested that at least 4OI interfered with SARS-CoV-2 infection in an NRF2-independent manner and that both 4OI and SEL interfered with viral entry. Indeed, in cell entry assays with a VSV vector pseudotyped with SARS-CoV-2 or SARS-CoV-1 spike protein (or VSV-G protein as positive control for ACE2-independent cell entry), 4OI and SEL (which enhances nuclear localization of ACE2[9]) markedly reduced entry of SARS-CoV-2 as well as SARS-CoV-1 (Fig. 2I), whereas reduction of SARS-CoV-2 cell entry by Bard was less pronounced (Supplementary Fig. S2A). Thus, both 4OI and SEL (and to a lesser extent Bard) interfered with SARS-CoV infectivity at the level of cell entry mediated by spike protein.

### Downregulation of ACE2, TMPRSS2, and XPO1

These results raised the hypothesis that the compounds affected expression of the main cellular receptor of SARS-CoV, angiotensin-converting enzyme 2 (ACE2), its functional co-receptor transmembrane serine protease 2 (TMPRSS2), or XPO1 (which reportedly regulates cytoplasmic localization of ACE2[9]). While SARS-CoV-2 infection did not alter expression of these mRNAs, the compounds markedly reduced levels of all three, in the order of efficiency 4OI > SFN = BARD (Fig. 3A–C). Immunoblotting verified that adding 4OI or SEL to uninfected Calu-3 cells gradually reduced ACE2 and TMPRSS2 protein levels, resulting in complete loss by 48 h (Fig. 3D), whereas ACE2 levels remained unchanged when the cells were cultured for 72 h in medium not containing the compounds (Supplementary Fig. S2B). We then performed a chase experiment with cycloheximide (CHX, which blocks translation) to test whether 4OI affects half-life of ACE2 protein. Addition of CHX to medium resulted in a notable reduction of ACE2 levels over 12 h (Fig. 3E), whereas adding CHX and 4OI together led to complete loss of ACE2 protein within 6 h. Densitometry allowed estimating the physiologic half-life of ACE2 (in the presence of CHX) to be about 12 h, whereas adding 4OI shortened this to much less than 3 h (Fig. 3F, G). Thus,

4OI dramatically reduced the half-life of ACE2 protein. ACE2 levels can be regulated by ubiquitination, leading to degradation by the proteasome[17]. We, therefore, tested the hypothesis that 4OI induces activity of the ubiquitin E3 ligases that contribute the most to ubiquitination, and thus proteasomal degradation, of ACE2. A network analysis of predicted E3 ligase targets of ACE2 revealed multiple interactions, with strongest functional relationships with MDM2 and NEDD4L (Supplementary Fig. S3). Indeed, knocking down NEDD4L mRNA with siRNA strongly reduced the ACE2-destroying capacity of 4OI (Fig. 3H). siRNA knock-down of MDM2 attenuated ACE2 destruction by 4OI as well, but the effect was less pronounced (Fig. 3I). We then used the proteasome inhibitors carfilzomib and MG132 to assess the role of proteasome in the observed ACE2 loss. Due to cytotoxicity at later time points (Supplementary Fig. S2E), only the 6 h time point could be tested. Surprisingly, both carfilzomib (Fig. 3J) and MG132 (Supplementary Fig. S2C, D) did not prevent 4OI-mediated loss of ACE2, but induced an even more rapid loss of ACE2 when given alone, suggesting that they induce ACE2 degradation by a different pathway. Indeed, inhibiting the proteasomal pathway has been shown to induce the lysosomal pathway of protein degradation[18]. Considering that polyubiquitinated proteins can also be funneled into the lysosomal pathway of protein degradation, we then tested whether lysosomal inhibitors would prevent ACE2 destruction by 4OI. Indeed, ACE2 levels were mostly preserved under chloroquine diphosphate and bafilomycin A1 treatment, strongly suggesting that 4OI leads to loss of ACE2 via the lysosomal pathway (Fig. 3K–N). Considering that 4OI treatment reduced XPO1 protein in uninfected and infected Calu-3 cells (Fig. 1E) and XPO1 mRNA in infected Calu-3 cells (Fig. 3C), we then compared the ability of 4OI and SEL to reduce XPO1 protein levels. XPO1 levels decreased greatly in the presence of either compound, with SEL exerting a somewhat stronger effect (Fig. 3O). Taken together, these results suggest that downregulation of ACE2, TMPRSS2, and XPO1 is a major component of the anti-SARS-CoV-2 effect of 4OI and SEL, and, to a lesser extent, also BARD and SFN.

### Reduction of ACE2 mRNA requires NEDD4L and MDM2 E3 ligase activity and correlates with reduced STAT3 phosphorylation

To explore whether 4OI- and SEL-mediated reduction in ACE2 mRNA and protein might share a mechanistic link, we tested whether the observed reduction in ACE2 mRNA also depended on NEDD4L and MDM2 E3 ligases. Indeed, siRNA knockdown of either ligase also mitigated reduction of ACE2 mRNA by 4OI and SEL (Fig. 3P, Q). STAT3 is a major transcription factor on the ACE2 promoter, and its cellular levels are in part regulated by polyubiquitination[19]. We, therefore, tested whether 4OI treatment would change levels of STAT3 or its phosphorylated active form. 4OI reduced levels of p-STAT3 but did not affect STAT3 levels (Fig. 3R). Thus, 4OI likely reduced transcription of the ACE2 promoter not by destabilizing STAT3 but by interfering with its phosphorylation.

### 4OI reduces XPO1 mRNA expression predominantly by repressing its transcription

Next, we investigated whether increased mRNA turnover played a role in the observed reduction of XPO1 mRNA by 4OI. This hypothesis was motivated by a report that siRNA-mediated knock-down of KEAP1 (leading to NRF2 activation) in human cells greatly shortens the half-life of STING mRNA, and addition of 4OI downregulates STING mRNA as well[12]. However, it is not known whether 4OI acts via the same post-transcriptional mechanism as KEAP1 inactivation. Considering that 4OI blocks nuclear export via XPO1[5], we performed a pathway analysis using XPO1 cargoes documented in the ValidNESs database[20] (Fig. 4A). Relevant enriched pathways included cellular functions involving macromolecule catabolism (autophagy), RNA turnover (mRNA surveillance, RNA degradation), but also RNA synthesis (RNA polymerase) and its activation (NF-κB signaling). Thus, XPO1 inhibition could affect mRNA levels of a given target by multiple mechanisms. We first tested the effect of 4OI on XPO1 mRNA half-life in uninfected A549 cells in the presence and absence of actinomycin D

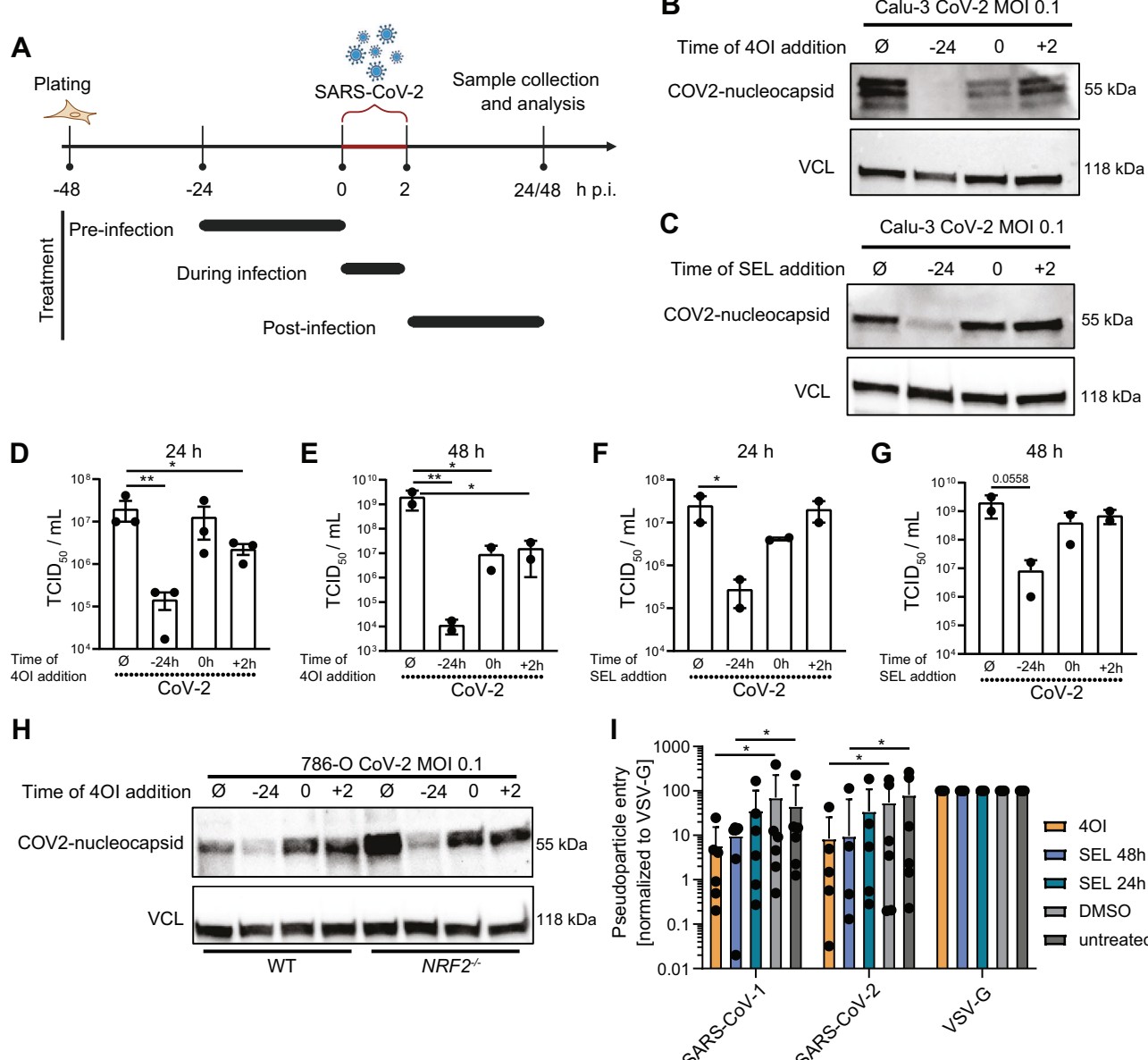

**Fig. 2 | Time of addition determines the anti-SARS-CoV-2 effect of the compounds. A–D** Calu-3 cells were treated with 4OI (100 μM) or SEL (1 μM) before, during, or after infection (inoculation with SARS-CoV-2/München-1.2/2020/984,p3, MOI = 0.1, for 2 h) as indicated in (**A**). Measurements were made 24 and 48 h p.i. Created in BioRender. Kalinke, U. (2026) https://BioRender.com/pzwv0ed. Immunoblots for SARS-CoV-2 nucleocapsid protein, 24 h p.i., treatment with 4OI (**B**) or SEL (**C**). **D–G** Viral titers (TCID$_{50}$/mL), 24 or 48 h p.i. **H** Experimental layout identical to **B** except that 786-O cells with CRISPR-mediated *NRF2*$^{-/-}$ were used (24 h p.i.). **I** 4OI and SEL interfere with cell entry of SARS-CoV-1 and -2. Calu-3 cells

were pre-incubated with 4OI (48 h) or SEL (24 or 48 h) and inoculated with luciferase-expressing VSV particles pseudotyped with SARS-CoV-1 or -2 spike protein or VSV-G protein. Cell entry was assessed by luciferase activity 16 h after inoculation. SARS-CoV pseudotype luciferase signal was normalized against the signal obtained with VSV-G pseudotypes, which was set as 100%. Pooled analysis of 6 independent experiments with $n = 3$ replicates each (**C**, **H**: $n = 2$). **D–G**: Lognormal two tailed *t*-test. **I** One-way ANOVA with Tukey's post-hoc test. * ≤0.05, ** ≤0.01, *** ≤0.001, **** ≤0.0001.

(ActD), which shuts down cellular transcription and allows measuring the natural decay of a given mRNA. *XPO1* mRNA levels decreased with similar kinetics when 4OI and ActD were applied as single treatments (Fig. 4C). If 4OI shortens *XPO1* mRNA half-life by a mechanism downstream of ActD, e.g., by increasing mRNA turnover, then the half-life should be shorter when both compounds are added together. Surprisingly, incubating the cells in medium containing both 4OI and ActD actually prolonged the half-life of *XPO1* mRNA (20.5 h vs. 10–12 h). The cytoprotective effects of 4OI are well documented and are largely mediated by induction of NRF2 signaling[21,22]. We hypothesized that 4OI diminished the effect of ActD via the efflux pump ABCB1, which is encoded by an NRF2 target gene and exports ActD from cells[23]. Indeed, when we knocked down *ABCB1* mRNA with siRNA

(Fig. 4B), the half-life-prolonging effect of 4OI/ActD co-treatment was greatly diminished, whereas the effect of ActD single treatment was enhanced, suggesting that 4OI attenuates the effect of ActD in an ABCB1-dependent manner (Fig. 4D). The hypothesis that 4OI and ActD act via a similar mechanism is also supported by differences between WT and knockdown cells when 4OI was applied after ActD. When WT cells (in which *ABCB1*-mediated export is expected to result in lower steady-state ActD levels) were treated for 24 h with ActD and then grown in fresh medium containing 4OI, *XPO1* mRNA levels continued to decline (Fig. 4C, right panel). On the other hand, in *ABCB1* knockdown cells (where ActD levels are expected to be higher and *XPO1* mRNA levels were indeed significantly lower at 24 h), a further reduction by the 4OI-containing medium was not

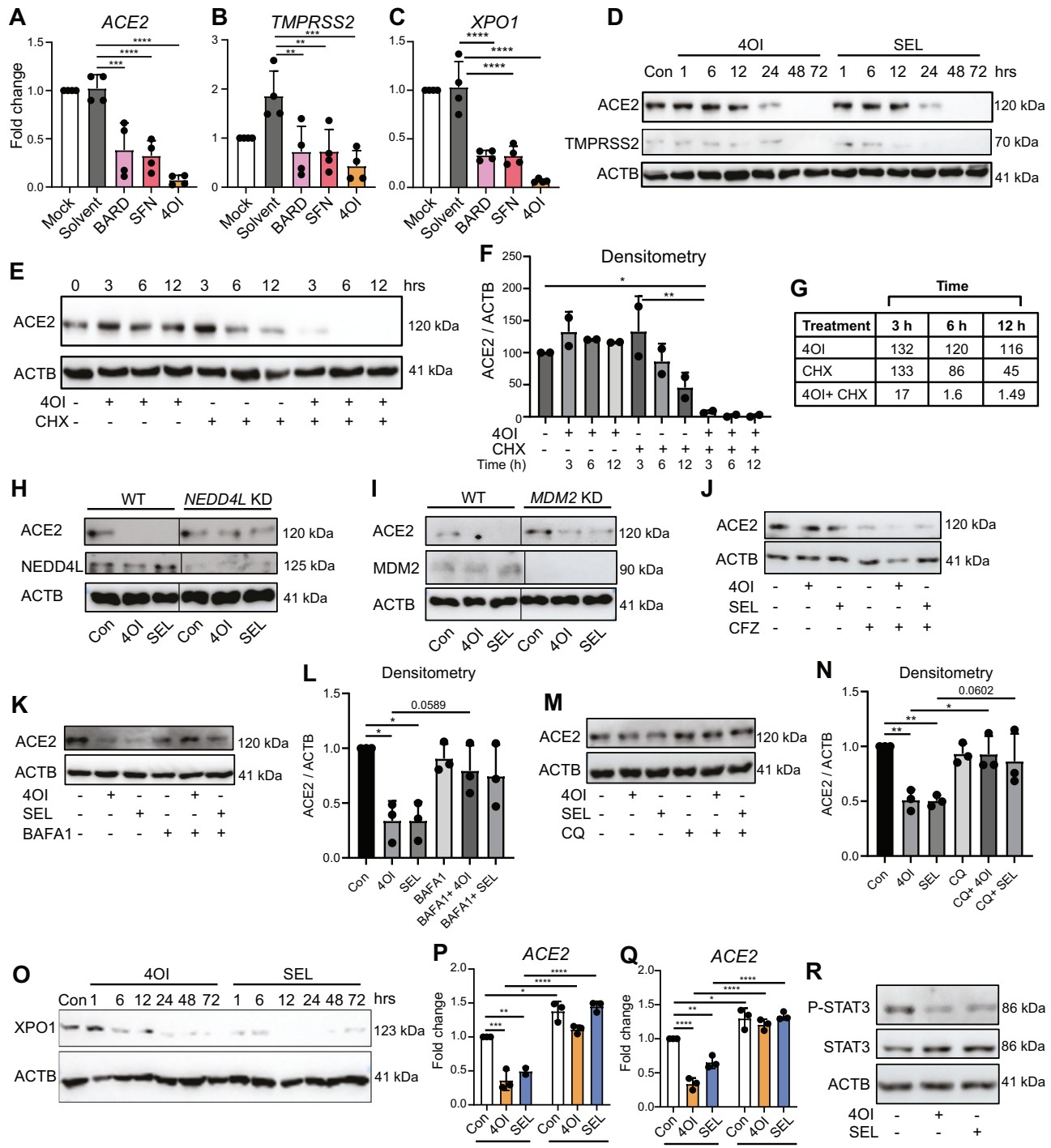

**Fig. 3 | 4OI and SEL reduce ACE2 and XPO1 levels. A–C** Reduction of *ACE2*, *TMPRSS2*, and *XPO1* mRNA. RT-qPCR analysis of RNA from the experiment shown in Fig. 1A–D. **D** 4OI and SEL reduce ACE2 and TMPRSS2 protein levels. Uninfected Calu-3 cells were grown in medium containing 4OI (100 μM) or SEL (1 μM), and cellular ACE2 and TMPRSS2 levels were measured by immunoblot after 1–72 h. **E** 4OI reduces half-life of ACE2. Uninfected Calu-3 cells were grown in medium containing 4OI with or without cycloheximide (CHX, 50 μg/ml), and cellular ACE2 levels were measured by immunoblot after 1–12 h. Densitometry of the blots shown in (**E**); **F**, bar chart; **G**, numerical values; untreated cells = 100%. **H** NEDD4L knock-down attenuates the ACE2-destroying capacity of 4OI at the protein. Efficiency of *NEDD4L* mRNA knock-down is shown in Supplementary Figure S2F. **I**. MDM2 knock-down attenuates the ACE2-destroying capacity of 4OI

at the protein level. Efficiency of *MDM2* mRNA knock-down is shown in Supplementary Figure S2G. **J–N** Uninfected Calu-3 cells were grown in medium containing 4OI, SEL, proteasome inhibitor (CFZ, 10 nM), or lysosome inhibitors (BAFA1-100 nM, CQ- 50 μM), and expression of ACE2 was determined by immunoblot. CFZ 6 h (**J**), BAFA1 24 h (**K, L**), and CQ 24 h (**M, N**). **O** 4OI and SEL reduce XPO1 protein levels. Uninfected Calu-3 cells were grown in medium containing 4OI or SEL, and XPO1 levels were measured by immunoblot after 1–72 h. **P, Q** NEDD4L and MDM2 knock-down attenuate ACE2 mRNA reduction by 4OI and SEL (RT-qPCR after 24 h). **R** 4OI and SEL attenuate STAT3 phosphorylation (immunoblot after 24 h). *n* = 3 (**D, E, H, I, O**: *n* = 2), means ± SEM. One-way ANOVA with Tukey's post-hoc test. * ≤0.05, ** ≤0.01, *** ≤0.001, **** ≤0.0001.

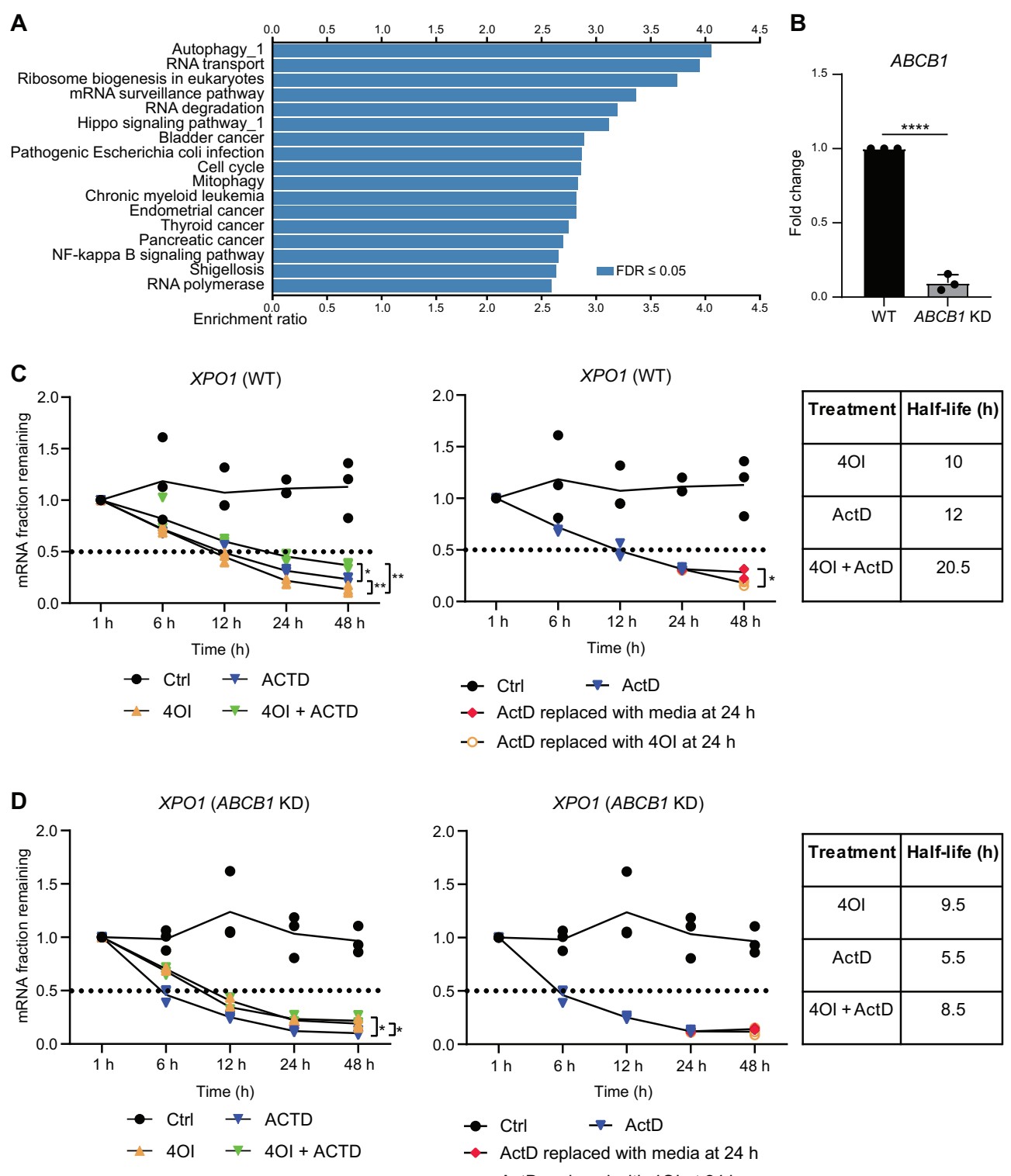

**Fig. 4 | 4OI reduces transcription of the *XPO1* gene. A** KEGG pathway analysis based on XPO1 cargoes listed in the ValidNESs database[20]. **B** Efficiency of *ABCB1* mRNA knock-down with siRNA (RT-qPCR). **C** Comparison of half-life reduction of *XPO1* mRNA by ActD, 4OI, or co-treatment with both. Numerical values for half-life were obtained by extrapolation and are listed in the table next to the graph. **D** Same experiment as in **C**, but using *ABCB1* knock-down (siRNA) A549 cells. Representative of two independent experiments, *n* = 3, means ± SEM. One-way ANOVA with Tukey's post-hoc test. * ≤0.05, ** ≤0.01, *** ≤0.001, **** ≤0.0001.

observed (Fig. 4D, right panel). Taken together, these results suggest that 4OI predominantly interferes with *XPO1* mRNA expression at the same step in gene expression as ActD, i.e., by inhibiting transcription. However, considering the potential confounding interactions between ActD and 4OI, we cannot fully exclude a contributing mechanism at the post-

transcriptional level. This would also agree with the close association of XPO1 function with RNA turnover found above (Fig. 4A). Nonetheless, in conjunction with the observed diminished STAT3 phosphorylation (Fig. 3R), we favor a model in which 4OI predominantly interferes with *XPO1* and *ACE2* mRNA expression by reducing their transcription.

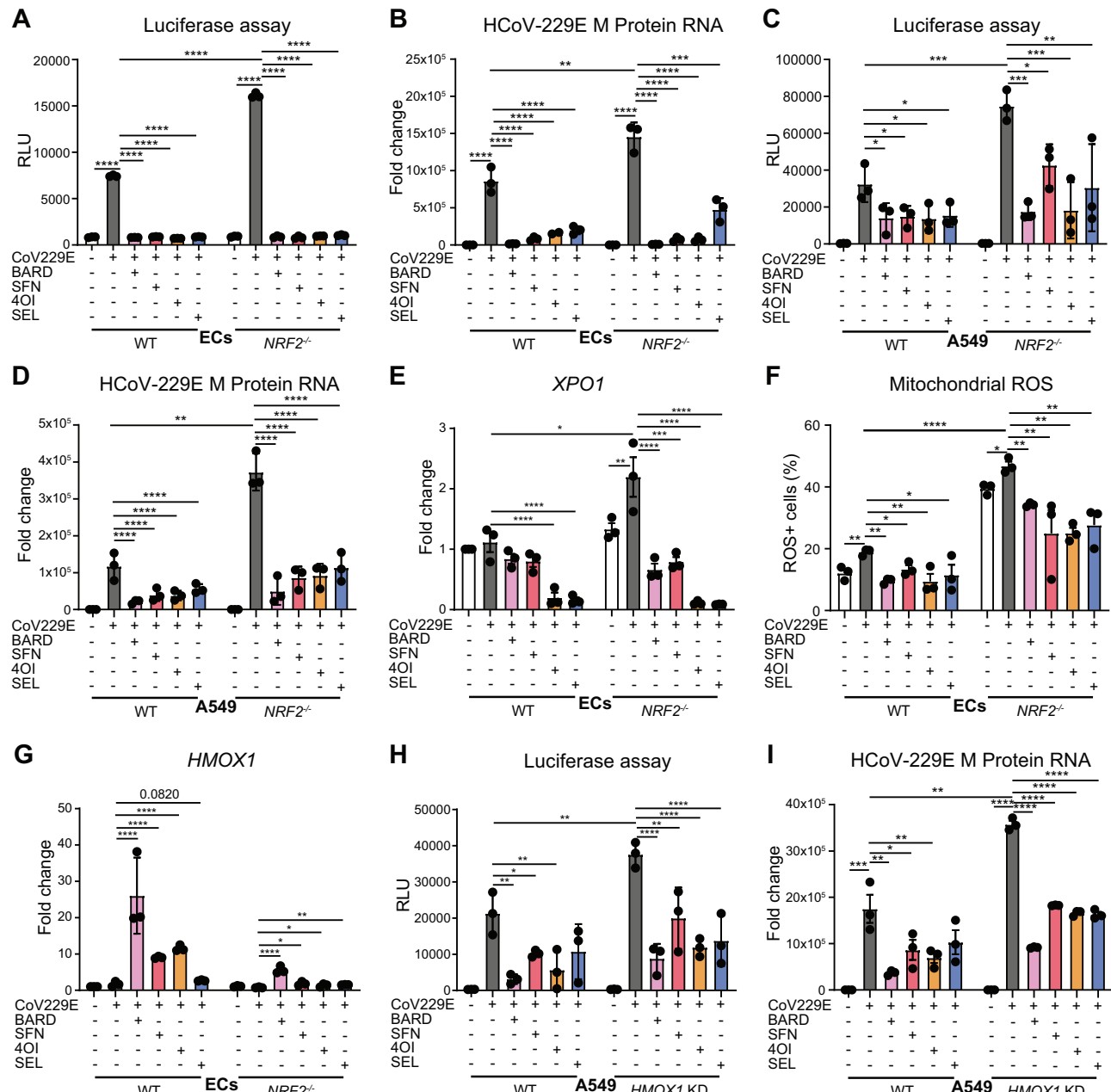

**Fig. 5 | NRF2-independent inhibition of 229E infectivity and downregulation of** *XPO1* **mRNA.** WT and *NRF2⁻/⁻* human iPSC-derived ECs (**A**, **B**, **E**–**G**) and A549 cells (**C**, **D**) were infected with the luciferase-labeled strain hCoV-229E-luc (MOI = 0.3) for 4 h and then cultured for 48 h in fresh medium containing the compounds. Luciferase activity, viral M protein RNA levels, host mRNA expression, and mitochondrial ROS levels were measured after 48 h. **A**, **B** Luciferase activity and viral M protein RNA. **C** Luciferase activity and (**D**) viral M protein RNA were measured in WT and *NRF2⁻/⁻* A549 cells. **E** *XPO1* mRNA. **F** Mitochondrial ROS (flow cytometry). **G** *HMOX1* mRNA. **H**, **I** Effect of HMOX1 knock-down on 229E-luc infectivity and antiviral activity of the compounds in A549 cells. Infections and treatments were carried out as in **A**–**G**, except that WT (transfected with scrambled control siRNA) or siRNA-mediated *HMOX1* knock-down A549 cells were used. **H** Luciferase activity. **I** M protein RNA. *n* = 3, means ± SEM. One-way ANOVA with Tukey's post-hoc test. * ≤0.05, ** ≤0.01, *** ≤0.001, **** ≤0.0001.

## NRF2-independent inhibition of seasonal coronavirus hCoV-229E

This low-pathogenic coronavirus can cause severe disease in immuno-compromised individuals. It shares cardinal features of viral replication with SARS-CoV-2, but can be studied under biosafety level 2 regulations. We therefore focused further studies on this pathogen. As opposed to SARS-CoV-2, hCoV-229E can productively infect microvascular endothelial cells (ECs)[24]. Indeed, a *Renilla* luciferase-encoding hCoV-229E strain (here referred to as 229E-luc) replicated efficiently in hiPSC-derived ECs, as evidenced by vigorous luciferase activity and high levels of RNA encoding the viral M protein (Fig. 5A, B). As observed with SARS-CoV-2 in 786-O

cells (see Fig. 2H), deleting the *NFE2L2* gene (which encodes NRF2) led to significantly increased viral replication, lending further support to its importance in restricting coronavirus infection in human cells. However, the three NRF2 activators (as well as SEL) prevented viral replication nearly completely in both WT and *NRF2⁻/⁻* ECs, indicating that their antiviral effect does not depend on NRF2 signaling. The NRF2 independence of the antiviral effects of the compounds was verified in A549 cells. This cell line supported higher Luc signals than the ECs (compare Fig. 5C vs. 5A), but the compounds were also somewhat less effective. Importantly, a similar reduction in viral replication was noted for all compounds in WT and *NRF2⁻/⁻* A549 cells (Fig. 5C, D). Of note, in contrast to *ACE2* and *TMPRSS2*

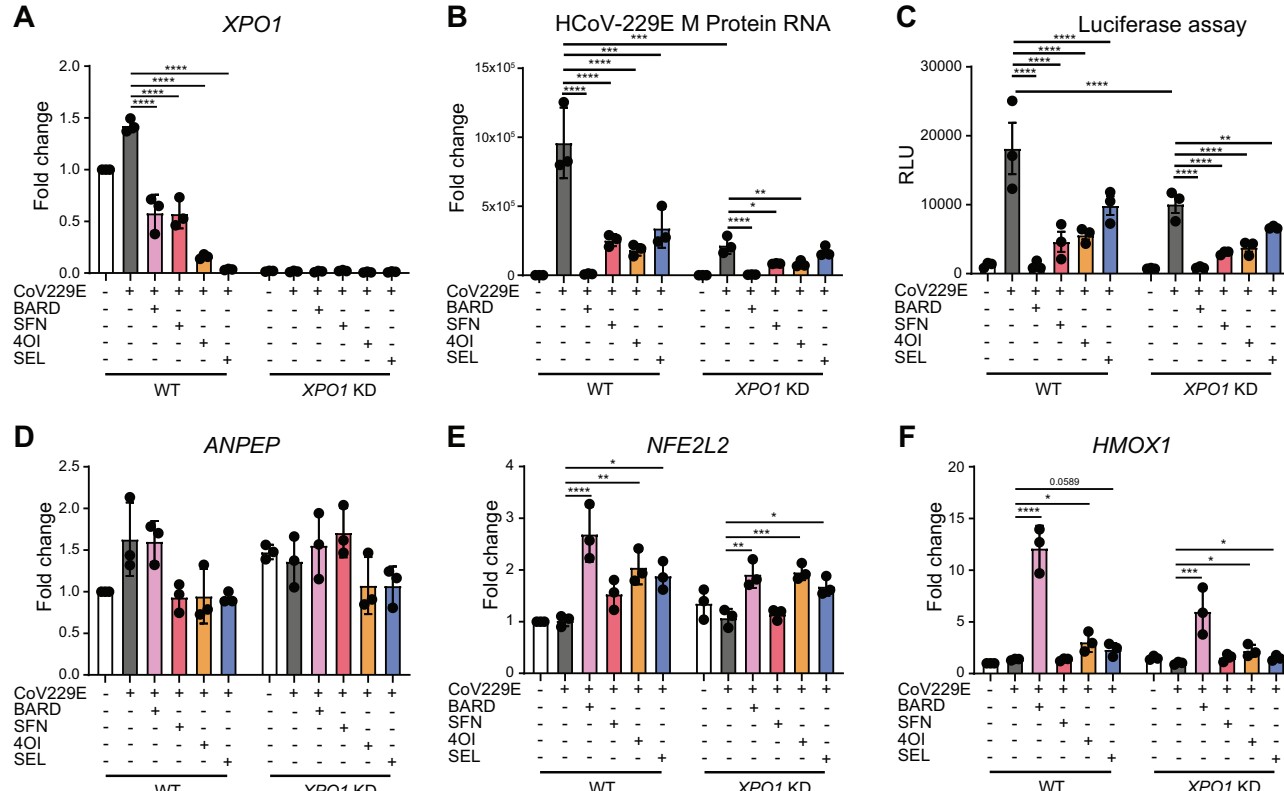

**Fig. 6 | XPO1 knock-down reduces infectivity of hCoV-229E.** *XPO1* mRNA was knocked down in A549 cells by siRNA, and infectivity of 229E-luc and antiviral efficacy of the compounds were assessed in WT (transfected with scrambled control siRNA) and XPO1 KD cells following the same infection protocol as in Fig. 3.

**A** *XPO1* mRNA (RT-qPCR). **B** M protein RNA expression. **C** Luciferase activity. **D** *ANPEP* mRNA (RT-qPCR). **E** *NFE2L2* mRNA (RT-qPCR). **F** *HMOX1* mRNA (RT-qPCR). $n = 3$, means ± SEM. One-way ANOVA with Tukey's post-hoc test. * ≤0.05, ** ≤0.01, *** ≤0.001, **** ≤0.0001.

mRNA in Calu-3 cells (see Fig. 3A, B), the compounds did not affect expression of mRNA encoding alanyl aminopeptidase (ANPEP), the major cellular receptor for hCoV-229E[25], in ECs, indicating that they do not restrict this virus by downregulating its cellular receptor (Supplementary Fig. S5G). However, as in Calu-3 cells, the three NRF2 activators reduced *XPO1* mRNA levels in ECs, with 4OI causing the greatest reduction (Fig. 5E). A similarly pronounced reduction was observed with SEL (Fig. 5E) and the naturally occurring XPO1 inhibitor leptomycin B (Supplementary Fig. S4), suggesting that the reduction in *XPO1* mRNA was mediated, at least indirectly, by binding of the compounds to XPO1 protein. In agreement with this, co-treatment of SEL in conjunction with 4OI, BARD, or SFN resulted in similar reduction of *XPO1* mRNA levels than treatment with SEL alone. *XPO1* mRNA levels were significantly higher in the *NRF2*$^{-/-}$ ECs, but the treatments reduced its expression to the same levels as in the treated WT ECs (Fig. 5E and Supplementary Fig. S4). Thus, downregulation of *XPO1* mRNA by the compounds did not require NRF2 signaling.

Levels of reactive oxygen species (ROS) were nearly 3-fold higher in uninfected *NRF2*$^{-/-}$ than WT ECs, underscoring the anti-oxidative impact of NRF2 signaling (Fig. 5F). 229E-luc infection led to a modest ROS increase in both *NRF2*$^{-/-}$ and WT cells, and the four treatments reduced ROS levels in the infected *NRF2*$^{-/-}$ cells below the level measured in the uninfected *NRF2*$^{-/-}$ cells. However, even though the three NRF2 activators had essentially abolished 229E-luc infection (Fig. 5A, B), ROS levels remained significantly higher in the treated *NRF2*$^{-/-}$ cells than in the treated WT cells, suggesting that the anti-oxidative effects of these compounds depended at least partially on NRF2 signaling. Indeed, the compounds induced the anti-oxidative NRF2 target gene *HMOX1* only weakly in the *NRF2*$^{-/-}$ cells (Fig. 5G). HMOX1 has been implicated as a downstream mediator of antiviral functions of NRF2[26]. We therefore assessed the impact of siRNA-mediated knock-down of *HMOX1* mRNA on 229E-luc infectivity (Fig. 5H, I). Considering the susceptibility of the iPSC-derived ECs to technical

artifacts from siRNA transfection, we performed this experiment in A549 cells. An approx. 80% knock-down of *HMOX1* mRNA was achieved (Supplementary Fig. S5J). Luciferase activity and M protein RNA expression were significantly higher in the *HMOX1* knock-down cells, supporting its antiviral role. Percent reduction of both parameters by the three NRF2 activators was only slightly lower in the *HMOX1* knock-down cells (Supplementary Table S1), suggesting that its contribution to the observed antiviral effect is relatively minor. This is also supported by the above results that full antiviral activity of the compounds was observed in the *NRF2*$^{-/-}$ ECs (Fig. 5A, B). SEL was somewhat more effective in the *HMOX1* knock-down cells, suggesting that HMOX1 interferes with its activity. This may relate to the previously described interaction between heme and XPO1-mediated nuclear export[27]. Neither *HMOX1* knock-down nor the treatments affected expression of *ANPEP* mRNA, which encodes the cellular receptor for hCoV-229E (Supplementary Fig. S5H).

## XPO1 is required for optimal infectivity of 229E and the antiviral effect of 4OI

Having demonstrated the NRF2-indepence of the compounds' anti-229E effects, we then assessed the contribution of XPO1 function to 229E infectivity and to their anti-229E activity. A 98% knockdown was achieved with anti-XPO1 siRNA in A549 cells, which was greater than the reduction seen after treatment with SEL (97%), 4OI (88%), Bard (59%), and SFN (59%) (Fig. 6A). The XPO1 knockdown led to a substantially larger reduction of M protein RNA (~80%) than of luciferase activity (~50%), which was comparable in both cases to the reduction by SEL treatment of WT cells (Fig. 6B, C). One can thus conclude that XPO1 contributes greatly to, but is not strictly required for, 229E infectivity in these A549 cells. As expected, SEL treatment of XPO1 knockdown cells did not reduce M protein RNA level or luciferase activity much further, confirming that, among the compounds tested herein, the antiviral activity of SEL depends the most

on the presence of XPO1. In contrast, a further reduction in these parameters ensued under treatment with BARD and, to a lesser extent, SFN and 4OI, suggesting that especially BARD acts at least partially through an additional target. A comparison of differences in percent reduction in 229E infectivity by the compounds between WT and knock-down cells suggested the following order of XPO1 dependence of the compounds: SEL > 4OI > SFN > BARD, which correlates well with their predicted affinities for the XPO1 NES binding site[5]. XPO1 knock-down did not affect expression of *ANPEP* mRNA, but it slightly reduced the ability of BARD to induce *NFE2L2* and *HMOX1* mRNA in 229E-infected cells, the functional relevance of which is uncertain (Fig. 6D–F).

### Changes in host cell transcriptomes due to hCoV-229E infection and treatment with 4OI, BARD, and SEL

We then used RNAseq to characterize the impact of 229E infection on host cell transcriptomes and to test whether the compounds differed in their effects on transcriptomes of infected cells. As expected, 229E infection of A549 cells led to major transcriptional reprogramming, both at 24 and 48 h p.i., which largely normalized upon treatment with all three compounds (Fig. 7A). For instance, 229E-luc infection led to a major increase in variance compared to mock-infected samples along both PC1 and 2 at 24 h, which increased further by 48 h, whereas the treated samples clustered close to the mock-infected samples at both time points. The Venn diagrams illustrate that there were differentially expressed (DE) genes unique to each compound and shared between two compounds, but that the largest numbers of DE genes (>5000 at both time points) were commonly DE upon treatment with all three compounds (Fig. 7B (48 h), Supplementary Fig. S6A (24 h)). In agreement with this, a gene set enrichment analysis (GSEA) based on KEGG pathways revealed that the three compounds affected very similar functional pathways in infected host cells (Fig. 7C (48 h), Supplementary Fig. S6B (24 h)). Targeted RT-qPCR analysis revealed that induction of the classic IFN-stimulated genes *IFIT1* and *CXCL10* was much lower in 229E infection than in SARS-CoV-2 infection (compare Supplementary Fig. S5 to Fig. 1C, D), which agreed with prior knowledge that differential gene expression in 229E-infected human cells does not feature a strong IFN response[28]. Indeed, the RNAseq analysis revealed that 229E infection resulted in minimal regulation of the 72 genes making up the KEGG pathway *Type I interferon response*: there were only 3 DE genes at 24 h p.i. (all up), and 5 at 48 h p.i. (2 up, 3 down) (Supplementary Fig. S6C and Fig. 7D). Treatment with BARD, 4OI, and SEL all reversed these expression changes. In contrast, infection led to a major downregulation of promitotic genes and upregulation of key antiproliferative genes such as *TP53* and *MYC*, which was remarkably reversed by treatment with all three compounds (Fig. 7E and Supplementary Fig. S6D). The weak IFN response was confirmed by the absence of IFN-related KEGG pathways in the GSEA comparing infected vs. uninfected cells (Fig. 8 (48 h), Supplementary Fig. S7 (24 h)). This was not due to an inability of A549 cells to support IFN responses, as infection with IAV leads to brisk IFN-responses in this cell type[29]. Rather, the GSEA revealed an overall cytostatic effect of the infection, as evidenced by significant depletion of pathways relating to cell cycle, nucleotide biosynthesis, and metabolism, which was reversed by treatment with the three compounds. The observation that the compounds normalized cell transcriptomes with such similar patterns suggests that the RNAseq analysis mostly reflects downstream effects of diminished viral replication and not specific modes of action of the compounds.

### Discussion

In this study of host-directed antivirals against high and low pathogenic human coronaviruses, we find that small molecules that activate the antiviral NRF2 signaling pathway greatly interfere with viral infectivity, but in an NRF2-independent manner. These results agree well with our previous investigations into the mode of action of the same compounds against influenza A virus, where we found that the antiviral activity did not depend on NRF2 signaling but was largely mediated by blocking the nuclear export factor XPO1[5], which was subsequently corroborated by Weiss et al.[15]. The

present study not only revealed the importance of XPO1 for infectivity of both SARS-CoV-1 and -2 and hCoV-229E, but it also suggests that XPO1 blockade mediates at least part of the anti-CoV activity of Nrf2 activators, in particular 4OI. This appeared counterintuitive because, as opposed to influenza viruses, coronaviruses complete the major phases of their life cycle in the cytoplasm. However, our findings lend further support to the notion that XPO1 blockade can target cytoplasmically replicating viruses by interfering with host cell factors required for the viral life cycle, in the case of SARS-CoV-1 and -2 by reducing availability of their cellular receptors ACE2 and TMPRSS2, as well as of XPO1 itself.

### Antiviral mode of action of 4OI and SEL

Olagnier et al. previously showed that 4OI treatment of SARS-CoV-2 infected cells relieved suppression of NRF2 signaling by the virus, but they did not provide experimental proof that this reconstitution of NRF2 signaling also mediated 4OI's antiviral effect. The present study now provides evidence that the anti-SARS-CoV-2 effect of 4OI is mediated to a large extent by downregulating ACE2, TMPRSS2, and XPO1 both at the protein and mRNA levels. Regarding the mRNA reduction, at least in the case of ACE2, it cannot alone account for the reduced protein level, as the CHX chase experiment clearly showed that 4OI acts on existing ACE2 protein by shortening its half-life several-fold. A similarly pronounced downregulation at the protein level was observed under SEL treatment, but it remains to be tested whether SEL and 4OI share a common mechanism of action. Cellular ACE2 levels are regulated by ubiquitination[30], including by the ubiquitin E3 ligases NEDD4L[31] and MDM2[32]. Indeed, both E3 ligases were necessary for 4OI to effect destruction of ACE2. In agreement with this, genetic ablation of ACOD1 (the enzyme that synthesizes itaconic acid in activated myeloid cells) leads to major alterations in cellular ubiquitination patters[33]. Surprisingly, we found that 4OI-mediated degradation of ACE2 did not require proteasomal activity but, rather, the lysosomal pathway. This agrees well with previous reports that 4OI can stimulate lysosomal biogenesis[34] and activate autophagy[35]. SEL induces ubiquitination and subsequent proteasomal degradation of XPO1[36], but we did not address the question whether it can alternatively target ACE2, TMPRSS2, and XPO1 via the lysosomal pathway. However, our pathway analysis of XPO1-associated cargos did identify *autophagy* as a significantly associated KEGG pathway. Even though we found that ACE2 destruction by 4OI largely required activity of at least two E3 ubiquitin ligases, we did not formally test whether treatment with these compounds actually led to hyperubiquitination of ACE2. Moreover, we did not test whether loss of TMPRSS2 required these (or other) E3 ligases as well. Clearly, further studies are required to verify these aspects of the presumed modes of action of the compounds.

Our results suggest that the observed diminished *ACE2*, *TMPRSS2*, and *XPO1* mRNAs levels are not due to changes in mRNA stability but, rather, by the compounds interfering with transcription of the respective promoters. In the case of *ACE2* mRNA, we could pinpoint diminished STAT3 phosphorylation as part of the mechanism, but it remains to be explained why knocking down the E3 ligases normalized levels of these mRNAs. There is no evidence that STAT3 activates the XPO1 promoter, but a plausible mechanism could entail enhanced nuclear localization of p53, which is a known silencer of the XPO1 promoter. Indeed, we previously found that blocking XPO1 with 4OI potentiates nuclear localization of p53 in respiratory epithelial cells[5]. Olagnier et al. had shown that activating NRF2 signaling by knocking down KEAP1 expression led to loss of *STING* mRNA by dramatically shortening its half-life, i.e., by a post-transcriptional mechanism[12]. In their work, addition of 4OI led to loss of *STING* mRNA in an NRF2-dependent manner, but it was not tested whether 4OI acted via the same post-transcriptional mechanism. In contrast, our ActD experiment (Fig. 4) suggests that 4OI largely acted by interfering with *XPO1* transcription. It has been reported that pharmacologically blocking XPO1 leads to loss of XPO1 protein[37–39], and *XPO1* mRNA levels have been measured under treatment with KPT-335 (verdinexor)[37] and KPT-185[38]. Intriguingly, expression was found to be increased, whereas in our study, treating A549

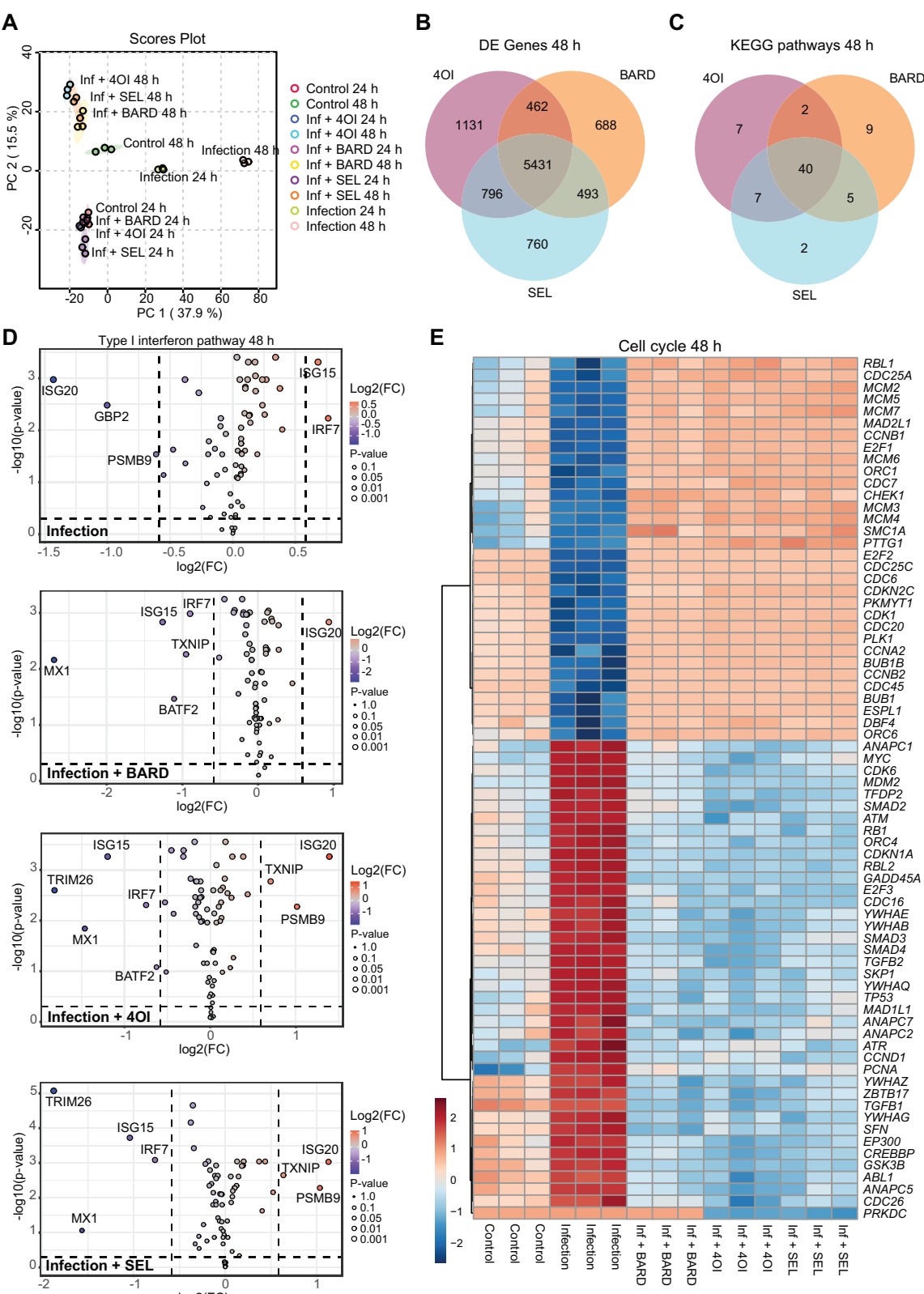

**Fig. 7 | Effect of BARD, 4OI, and SEL on transcriptomes of 229E-infected A549 cells.** A549 cells were infected with 229E-luc, treated with the compounds as indicated, and cell transcriptomes were determined by RNAseq 24 and 48 h p.i. **A** PCA showing centroids of values obtained from the 3 samples in each group. Controls = mock-infected samples. Venn diagrams indicating shared and unique DE genes (**B**) and KEGG pathways (**C**) 48 h p.i. **D** Differential expression of the genes contained in KEGG pathway *Type I interferon pathway* due to 229-luc infection without treatment and under treatment with BARD, 4OI, and SEL. **E** Hierarchical clustering analysis (*y*-axis) of 72 of the 113 genes contained in KEGG pathway *Cell cycle*, which were selected by highest |log2FC| values. The color scale on the lower left indicates *Z*-score.

**Fig. 8 | hCoV-229E infection represses signaling pathways associated with cell proliferation and metabolism, which is reversed by treatment with BARD, 4OI, and SEL.** GSEA (KEGG pathways) based on the RNAseq data 48 h p.i. used for Fig. 6. KEGG pathways were selected by pAdj <0.05 and are listed in reverse alphabetical order top to bottom. NES, normalized enrichment ratio.

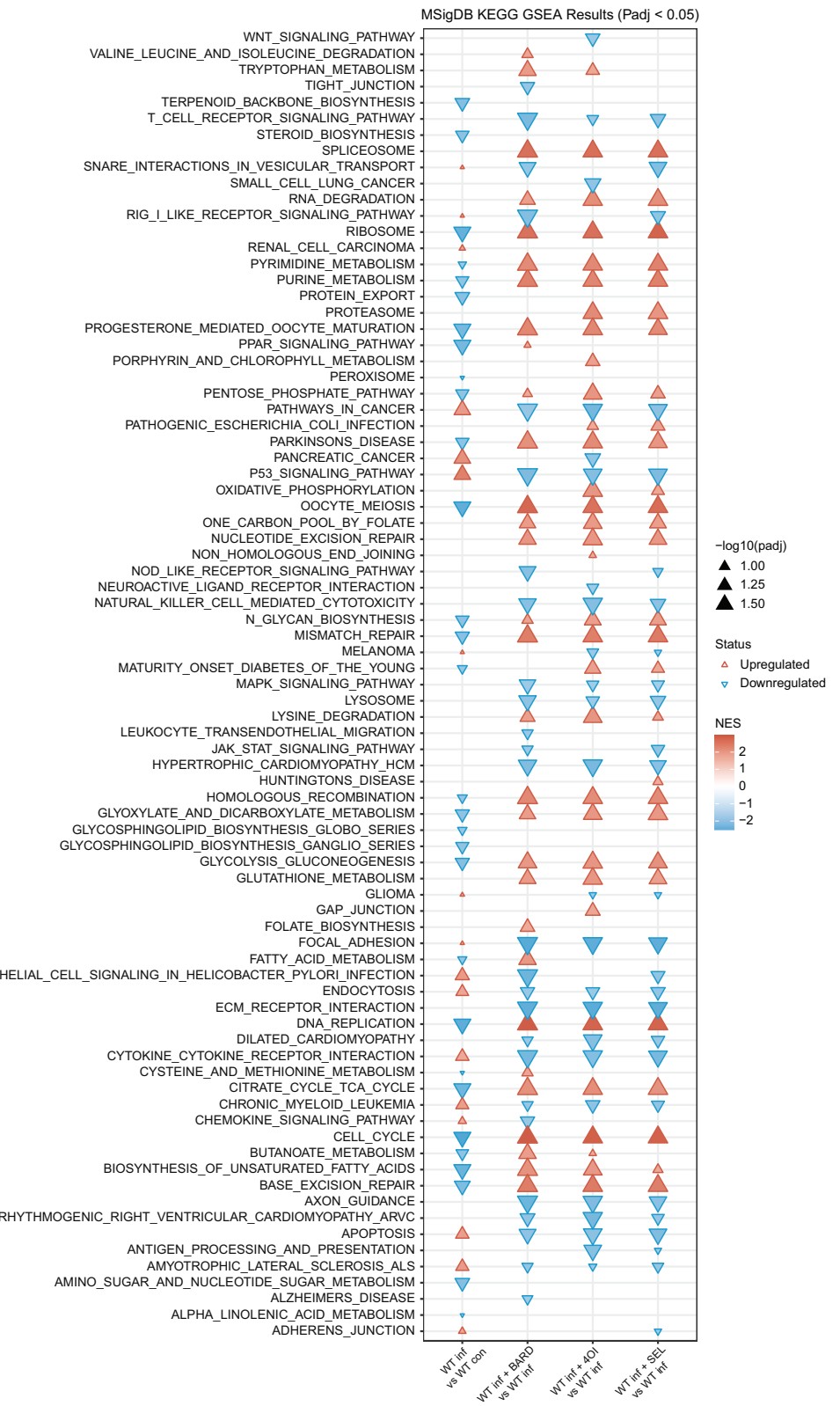

and Calu-3 cells with SEL (KPT-330) clearly reduced *XPO1* mRNA levels. Thus, the mechanism underlying the loss of XPO1 protein may differ depending on factors such as the pharmacologic agent and cell type studied. In any case, 4OI downregulated *XPO1* mRNA more efficiently than SEL, suggesting that it interferes with XPO1 expression also through an XPO1-independent mechanism. Indeed, recent work revealed inhibition of TET2 dioxygenase (an enzyme that tends to activate gene transcription by reversing promoter methylation) as a major mechanism of the anti-inflammatory effect of 4OI and endogenous itaconic acid[40]. Other plausible mechanisms include interference with NF-κB activity, for instance by preventing export of IκB-α from the nucleus, as has been described for SEL in breast cancer cells[41].

Olagnier et al. showed that 4OI reduces HSV1/2 infectivity as well, but that this effect is at least partially NRF2-dependent. This suggests that induction of antiviral NRF2 target genes plays a more prominent role in the antiviral mode of action of 4OI against herpes viruses. Indeed, pharmacologic induction of HMOX1 has been shown to interfere with HSV-2 replication in epithelial cells and neurons[42].

The mechanism underlying 4OI's anti-229E activity is only partially explained by our results. It appeared to be partially independent of XPO1, and it did not affect expression of its cellular receptor, ANPEP (Supplementary Fig. S5), suggesting the presence of other targets. On the other hand, the anti-229E activity of SEL required XPO1, suggesting that XPO1 mediates an important step in the 229E life cycle. Several SARS-CoV-2 proteins are potential XPO1 cargos. ORF6, Nsp6, ORF7a, and NSP7, interact with nuclear cytoplasmic transport, whereby ORF6 interacts the most strongly with XPO1[43]. Of these, the 229E genome encodes only one ortholog (Nsp6)[44], and it is not known whether XPO1 transports CoV-229E polypeptides in a nuclear-cytoplasmic shuttle. Further research is, therefore, required to clarify XPO1-dependent and -independent mechanisms of 4OI's and SEL's anti-CoV229E activity.

Our study primarily aimed to elucidate the mode of action of the compounds, and the results were thus obtained in cellular models. However, the in vivo relevance of the anti-SARS-CoV-2 activity of SEL (which, according to our results, inhibits SARS-CoV-2 largely by the same mechanism as 4OI, but somewhat less efficiently) was demonstrated in a ferret model of SARS-CoV-2 infection[9]. Specifically, oral treatment with SEL (5 mg/Kg twice daily for 3 days) led to lower viral burden and lung tissue damage and reduced expression of pro-inflammatory cytokines after infection with the BetaCoV/Munich/BavPat1/2020 strain.

### Antiviral mode of action of BARD
Antiviral efficacy of 4OI and BARD against SARS-CoV-2 has previously been demonstrated in cellular models[6,45]. Sun et al. suggested that BARD may have direct anti-SARS-CoV-2 activity, presumably by binding to its 3C-like protease[45]. However, the half-maximal effective concentration (EC$_{50}$) to inhibit SARS-CoV-2 replication in cellular infection models was substantially lower than the half-maximal inhibitory concentration (IC$_{50}$) to inhibit the enzyme in a cell-free assay, suggesting the presence of an additional, likely cellular, target. We find that BARD interferes with both SARS-CoV-2 and hCoV-229E. The latter expresses a 3c-like protease with a similar structure, but it is not known whether it can be inhibited by BARD. BARD did reduce expression of ACE2 and TMPRSS2, but the effect was relatively modest, and it inhibited SARS-CoV-2 spike protein-mediated cell entry less than 4OI. In addition, its XPO1 blocking activity is markedly lower than that of 4OI or SEL[5]. The anti-229E activity of BARD did not require NRF2, was less XPO1-dependent, and did not affect ANPEP expression, suggesting either a direct antiviral effect or the presence of another cellular target. It has been suggested that BARD inhibits the PI3K/Akt/mTOR and p38 MAPK signaling pathways[46]. These pathways are induced by a variety of viral pathogens, including coronaviruses[47,48] and influenza A virus[7,49], and are proposed targets for host-directed antivirals. However, inhibiting these pathways by BARD induced cell cycle arrest in K562 cells, i.e., the opposite of what was observed by us in 229E-infected A549 cells. Moreover, the GSEA analysis did not reveal regulation of these pathways by 229E infection or BARD treatment. Thus, a cellular target(s) which mediate(s) the presently unexplained portion of BARD's anti-CoV effect remains to be identified.

### Importance of endogenous NRF2 signaling
The NRF2 signaling pathway has been extensively implicated in host defenses against pathogenic viruses, and HMOX1 is considered a downstream mediator of this antiviral effect[26]. Our results underscore the strong anti-oxidative function of endogenous NRF2 signaling even in uninfected cells. Moreover, they confirm its antiviral capacity, as replication of both SARS-CoV-2 and 229E increased upon inactivation of the *NFE2L2* gene (Figs. 2H and 5D), which was paralleled by a decrease in *HMOX1* mRNA expression. Olagnier et al. showed that knocking down *HMOX1* mRNA

expression did not affect SARS-CoV-2 infection or replication in African Green Monkey Vero cells[6], whereas in our hands knocking down *HMOX1* mRNA enhanced 229E infection. It appears puzzling that the reduced *HMOX1* expression in the NRF2 KO ECs allowed full antiviral activity of the compounds whereas knocking down *HMOX1* in A549 cells diminished the anti-229E effect of the compounds, albeit in a relatively minor way. The activity of HMOX1 against human coronaviruses may thus vary depending on virus and/or cell type and host species.

In summary, we find that the anti-CoV activity of the NRF2-activating compounds 4OI, BARD, and SFN (as well as the XPO1 inhibitor SEL, which is also predicted to bind to the NRF2 inhibitor KEAP1[5]) does not require an intact NRF2 signaling pathway and that a good part of the mode of action of 4OI and SEL, and to a lesser extent also BARD and SFN, is mediated by downregulating ACE2 and TMPRSS2 and blocking/downregulating XPO1. The latter agrees well with our previous ligand-target modelling results, which suggested that these NRF2 activators and SEL can recognize each other's canonical binding sites (KEAP1 and XPO1) due to the presence of cysteines, which are displayed in similar environments that are conducive to formation of covalent bonds with electrophilic double bonds in the compounds[5]. The higher affinity of 4OI, compared with BARD and SFN, for XPO1 was explained by additional hydrophobic interactions of its 4-octyl tail with the hydrophobic cleft of XPO1. Considering the ability of electrophilic compounds to interact with a multitude of nucleophilic targets in the cell, their complete mode of antiviral action likely results from the sum of diverse additive, counteractive, and synergistic interactions, which may vary according to cell type, activation status, and life cycle of the virus. Considering that they integrate antiviral, anti-inflammatory, and cytoprotective properties, electrophilic NRF2-activating small molecules like 4OI, SFN, and BARD certainly constitute promising starting points for the development of host-directed antivirals.

## Methods and materials
### Compounds
Bardoxolone methyl (BARD; 2-cyano-3,12-dioxo-oleana-1,9(11)-dien-28-oic acid methyl ester, Hölzel/MedChem Express, Sollentuna, Sweden, HY-13324-50mg), 4-octyl itaconate (4OI; Biomol Cayman chemical, Hamburg, Germany, Cay25374-25), sulforaphane (SFN; (R)-1-isothiocyanato-4-(methylsulfinyl)-butane; Santa Cruz, Heidelberg, Germany, sc-203099), selinexor (SEL; selective inhibitor of nuclear export KPT-330, Tebu-Bio, Offenbach, Germany, 10-4011-25 mg). We used previously documented nontoxic concentrations which had antiviral activity: 4OI 100 µM[7], SEL 1 µM[50], BARD 0.1 µM[51], SFN 10 µM[52]. The lack of toxicity of the compounds at these concentrations on Calu-3 cells was additionally verified by LDH release assay (Supplementary Fig. S1). Leptomycin B (LMB, 431050-5UG, Sigma Aldrich) was applied at a concentration of 20 nM, and MG132 (S2619, Hölzel Diagnostica) at 10 µM. Carfilzomib (CFZ, PR171, Hölzel diagnostika) was used at a concentration of 10 nM, bafilomycin A1 (BAFA1, S1413-1, Hölzel diagnostika) at 100 nM, and chloroquine diphosphate (CQ, C6628, Sigma-Aldrich) at 50 µM.

### Cells and cell lines
Calu-3 epithelial lung cancer cells were kindly provided by Laureano de le Vega, Dundee University, Scotland, UK. Cells were grown in Dulbecco's Modified Eagle Medium (DMEM) (ThermoFisher) supplemented with 10% heat-inactivated FCS (Sigma-Aldrich), 1000 U ml−1 penicillin (Gibco), 1000 µg ml−1 streptomycin (Gibco), and 2 mM L-glutamine (Gibco). 786-O cells were purchased from ATCC. The stable NRF2 KO 786-O cells used in this study were previously generated using a CRISPR/Cas9 approach[16]. Cells were transfected with the pLentiCRISPR-v2 vector containing sgRNA targeting the KEAP1-binding domain of the *NFE2L2* gene. After puromycin selection, single-cell clones were isolated and validated. Control 786-O cells correspond to a pooled population transfected with an empty pLentiCRISPR-v2 vector and subjected to the same selection procedure[16]. A549 lung adenocarcinoma cells were originally obtained from German Collection of Microorganisms and Cell Cultures GmbH (DSMZ,

Braunschweig, Germany) and were grown in DMEM medium supplemented with 10% FCS and 2 mM L-glutamine. A549pLenti and NRF2 KO cells were provided by Laureano de la Vega (University of Dundee, Scotland) and have described previously[53]. Huh-7.5 and HEK293T cells were cultured in Dulbecco's modified Eagle's medium supplemented with 10% fetal calf serum, 2 mM L-glutamine, 0.1 mM nonessential amino acids, and 1% penicillin/streptomycin at 37 °C and 5% $CO_2$. Generation and verification of $NRF2^{-/-}$ iPSC was described in ref. 54, and the absence of NRF2 gene expression was verified again by RT-qPCR (Supplementary Fig. S5I). Isogenic WT and $NRF2^{-/-}$ iPSC cells[54] were differentiated into CD31+ vascular ECs using an established protocol[55] and cultured in ECGM-2 medium (PromoCell, Darmstadt, Germany) on plates coated with fibronectin (Corning, NY, USA).

## Methods (Aarhus, Figs. 1E–G and 2A–H)

**SARS-CoV-2 and variants of concern.** SARS-CoV-2 Wuhan-like early European B.1 lineage (FR-4286) was kindly provided by Professor Georg Kochs (University of Freiburg). Professor Arvind Patel (University of Glasgow, UK) kindly provided the alpha variant B.1.1.7, and the beta variant B.1.351 was kindly provided by Professor Alex Sigal, African Health Research Institute, South Africa. Delta variant B.1.617.2 (SARS-CoV2/hu/DK/SSI-H11) was provided by Statens Serum Institut, SSI, Denmark. The viruses were propagated as previously described[56]. Briefly, $10 \times 106$ VerohTMPRSS2 cells were seeded in T175 cell culture flasks and infected the following day at an MOI of 0.005 in 10 mL of cell culture media containing serum. After 1 h, the cell culture medium was increased to 20 mL per flask, and virus propagation continued for 72 h. Cell debris was removed by centrifugation of the cell culture supernatants from the different flasks at 300 g for 5 min. Viruses were concentrated in Amicon filter tubes by spinning at 4000 g for 30 min at 4 °C. The concentrated virus was further aliquoted and stored at −80 °C. The amount of infectious virus was determined using an endpoint dilution assay ($TCID_{50}$) as described below.

**SARS-CoV-2 infection experiments.** Calu-3, 786-O WT, and NRF2-KO cells were seeded in 24-well plates ($2 \times 10^5$ and $1.5 \times 10^5$ cells/well, respectively), allowed to adhere overnight, and infected the following day with SARS-CoV-2 at a multiplicity of infection (MOI) of 0.1. Supernatants were collected at 24 h and 48 h post-infection for quantification of viral replication by $TCID_{50}$ assay. Cell lysates were harvested at 24 h post-infection to assess viral protein expression by immunoblotting.

**Real-time quantitative reverse transcriptase polymerase chain reaction (RT-qPCR) for SARS-CoV-2.** Cells were washed with PBS and lysed in 300 μL of RNA lysis buffer (Roche) diluted with 200 μL of PBS, followed by RNA extraction and gene expression analysis. RNA was extracted using the High Pure RNA Isolation Kit (Roche) according to the manufacturer's instructions. RNA samples were diluted to 100 ng/uL and gene expression was analyzed by real-time quantitative PCR using TaqMan® RNA-to-CT ™ 1-Step Kit (Applied Biosystems) as previously described, whereby the assay for viral RNA detects predominantly genomic RNA with a possible contribution of subgenomic RNA[56]. Primers and probe sequences, hybridizing to the nucleocapsid region, were provided by the CDC and purchased from Eurofins. Samples were analyzed in a final volume of 10 μL, containing 5 μL of master mix, 0.5 μL (10 pmol/μL) of forward primer, 0.7 μL of reverse primer (10 pmol/μL), 0.2 μL of probe (20 pmol/μL), 2.4 μL of nuclease-free water, and 1 μL of diluted RNA. The analysis was performed on a ThermoFisher Scientific qPCR machine (Quant Studio 5). mRNA encoding TATA-Box Binding Protein (TBP) was used as internal control and primers and probe were obtained from ThermoFisher.

**LDH release assay.** Cellular toxicity of the respective compounds was measured 48 h post-treatment using CyQUANT™ LDH Cytotoxicity Assay Kit (ThermoFisher) according to the manufacturer's instructions.

Untreated cells were used as a negative control, whereas cells lysed with the provided lysis buffer served as a positive control. LDH activity was determined following subtraction of the background 690 nm absorbance value from the 450 nm absorbance value measured on a BioTek Microplate Reader (BioTek Instruments). The percentage of cytotoxicity was calculated using the following formula: (LDH activity – LDH activity control)/(LDH activity positive control – LDH activity negative control) × 100.

**$TCID_{50}$ assay.** The compounds 4-octyl itaconate (4OI, 100 μM) and selinexor (1 μM) were added 24 h prior to infection, at the time of infection, or 2 h post-infection. Supernatants from infected cultures were serially diluted 10-fold (from $10^{-1}$ to $10^{-10}$) in DMEM. Vero E6-TMPRSS2 cells were seeded in flat-bottom 96-well plates at $2 \times 10^4$ cells/well one day prior to infection. For each sample and each dilution, four technical replicates were used. Cells were infected with the serial dilutions and monitored for cytopathic effect (CPE) daily. CPE was recorded at 4 days post-infection using an inverted microscope. Viral titers were calculated using the Reed–Muench method, and results were expressed as $TCID_{50}$/mL. All experiments were performed with at least three independent biological replicates.

**Immunoblotting.** Immunoblotting was performed as described[6]. Cell lysates were harvested at 24 h post-infection to assess viral protein expression by immunoblotting. Briefly, cells were washed with PBS and subsequently lysed in 100 μL ice-cold Pierce RIPA lysis buffer (ThermoFisher Scientific), supplemented with 10 mM NaF, 1x complete protease inhibitor cocktail (Roche), and 5 IU/mL Benzonase Nuclease (Sigma). Protein concentration was measured using a BCA Protein Assay Kit (ThermoFisher Scientific). Whole-cell lysates were resuspended in a loading buffer, consisting of XT Sample Buffer (BioRad) and XT Reducing Agent (BioRad). Samples were then denatured at 95 °C for 5 minutes (min). 10–40 μg of the reduced sample was separated by SDS-PAGE on a 4–20% Criterion TGX pre-cast gradient gel (BioRad). Each gel was first run at 70 V for 20 min, following 110 V for 50 min. Proteins were transfer onto a Midi Format 0,2 μM PVDF membrane (BioRad) using a Transblot Turbo Transfer System (BioRad) for 7 min. Membranes were blocked for 1 h at room temperature with 5% skim milk (Sigma–Aldrich) in PBS supplemented with 0.05% Tween-20 (PBS-T). Membranes were fractionated into smaller pieces and incubated overnight at 4 °C with one of the following primary antibodies in PBS-T and 0.02% sodium azide: mouse mAb anti-vinculin (Sigma #V9264, 1:10.000), mouse IgG1 anti-SARS-CoV Spike (Genetex #GTX632604, 1:1000), Rabbit anti-SARS-CoV2 Nucleocapsid (Cell Signaling Technology #26369, 1:1000), Rabbit anti-Exportin1-XPO1 (Cell Signaling Technology #46249, 1:1000), mouse anti-AKR1B10 (Santa Cruz #SC-365689, 1:1000), rabbit anti-NQO1 (Cell Signaling Technology #62262, 1:1000), and rabbit anti-Beta-actin - ACTB (Cell Signaling Technology #8457, 1:1000) was used as a loading control. Membranes were washed 3 times in PBST for 15 min following incubation with secondary antibodies: donkey anti-mouse IgG (H + L) (Jackson Immuno Research #715-036-150), donkey anti-rabbit IgG (H + L) (Jackson Immuno Research # 711-035-152) in PBS-T 1% skimmed-milk for 1 h at RT. Membranes were washed 3 times in PBS-T for 10 min and subsequently incubated with SuperSignal West Dura Substrate or SuperSignal West Femto Maximum Sensitivity Substrate (ThermoFisher Scientific) for 1 min prior to exposure using iBright CL1500 Imaging System (ThermoFisher Scientific).

## Methods (Hannover)

**Viruses.** SARS-CoV-2 infection assays were carried out using strain SARS-CoV-2/München-1.2/2020/984,p3) (Wölfel et al. 2020), kindly provided by Christian Drosten (Charité, Berlin) through the European Virus Archive – Global (EVAg). After primary isolation from the patient, the virus was propagated in and titrated on Vero cells (passage 3). The recombinant HCoV-229E encoding a *Renilla* luciferase gene (kindly

provided by Volker Thiel, University of Bern, Switzerland[57], here referred to as 229E-luc) was grown at 33 °C on Huh7.5 cells and titrated in the same cell line.

**Infection assays.** For SARS-CoV-2 infection, $4.5 \times 10^5$ Calu-3 cells/well were seeded in 24-well plates coated with collagen. The next day, cells were pretreated with the indicated compounds for 24 h at 37 °C and 5% $CO_2$ and then inoculated with the SARS-CoV-2 isolate (MOI 0.005) in presence of the compounds and controls. Heat-inactivated virus (15 min at 70 °C) served as input viral RNA (vRNA) control. The inoculum was removed after 4 h, and cells were washed twice with 1x PBS and fresh medium complemented with compounds. At 48 h p.i., vRNA was isolated from supernatants (NucleoSpin RNA kit, Macherey Nagel) and cell lysates (QIAamp Viral RNA Mini Kit, QiaGen), and viral genome copies were determined by RT-qPCR using SuperScriptIII one-step RT-PCR and Platinum Taq Polymerase (Invitrogen)[58]. vRNA amplification methodology, using a specific in vitro-transcribed RNA quantification standard, was described previously[59]. For HCoV-229E-luc infections of iPSC-derived ECs and A549 cells, cells were seeded in 12- or 96-well plates and inoculated with a luciferase-labeled HCoV-229E-luc at an MOI of 0.3 for 4 h. After removing the viral inoculum, fresh cell culture media complemented with compounds were added, and the cells were collected 48 h.p.i. Either for RT-qPCR to determine viral genome copies using SuperScript™ III One-Step RT-PCR System Platinum™ Taq DNA Polymerase (Thermo Fisher Scientific) or for *Renilla* luciferase activity to determine infectivity and replication.

**VSV pseudoparticle production and cell entry assay.** Recombinant VSV pseudoparticles bearing SARS-CoV-2, SARS-CoV-1, or VSV glycoproteins were generated as described[60,61]. In short, HEK293T cells were transfected with vectors expressing SARS-CoV-2-S, SARS-CoV-1-S, VSV-G, or with empty expression vector as negative control. After 24 h, cells were inoculated for 1 h with a replication-deficient VSV vector containing a Firefly luciferase and an eGFP expression cassette instead of the VSV glycoprotein (kindly provided by Gert Zimmer, Institute of Virology and Immunology, Switzerland). Subsequently, the inoculum was removed, the cells were washed twice with 1x PBS, and medium was supplemented with anti-VSV-G antibodies (I1, mouse hybridoma supernatant from CRL-2700) added, except to those viruses expressing VSV-G. Cell culture supernatant was harvested 16 h post-inoculation and clarified from cell debris by centrifugation ($2000 \times g$ for 10 min) prior storage at −80 °C. For cell entry assays, $5 \times 10^4$ Calu-3 cells/well were seeded on collagen-coated 96-well plates. The next day, cells were pre-treated with the indicated compound for either 24 h or 48 h, refreshing compounds after 24 h for the latter. After pretreatment, cells were washed with 1x PBS and inoculated with 50 µl of the indicated pseudoparticles in the absence of the compounds. Ninety minute post inoculation, 150 µl of complete media was added onto the inoculum. At 16 h post inoculation, cells were washed once with 1x PBS and lysed with 50 µL/well of Luc-lysis buffer (1% Triton-X 100, 25 mM Gly-Gly (pH 7.8), 15 mM MgSO4, 4 mM EGTA, 1 mM DTT in $H_2O$). Transduction efficiency was measured by luciferase activity using a microplate reader (Berthold Technologies). To show SARS-CoV-1 and SARS-CoV-2 spike-specific entry effects, the background for each treatment was subtracted (values from the negative control; no env pseudoparticles), and data were normalized to the signal obtained with VSV-G.

**Gene knock-down with siRNA.** For XPO1 knock-down, cells were grown to 90% confluency and transfected with specific (ON-TARGET-plus Human XPO1 (7514) siRNA—SMARTpool, 5 nmol, L-003030-00-0005, Horizon Discovery) or control siRNA (ON-TARGETplus Non-targeting Control Pool, D-001810-10-05, Horizon Discovery) using Opti-MEM medium (31985070, Gibco). ON-TARGETplus Human ABCB1 siRNA—SMARTpool, 5 nmol (L-003868-00-0005, Horizon Discovery) was used to knock down *ABCB1* mRNA. ON-TARGETplus

Human HMOX1 siRNA—SMARTpool, 5 nmol (L-006372-00-0005, Horizon Discovery) was used to knock down *HMOX1* mRNA. ON-TARGETplus Human NEDD4L siRNA—SMARTpool, 5 nmol (L-007187-00-0005, Horizon Discovery) was used to knock down *NEDD4L* mRNA, and ON-TARGETplus Human MDM2 siRNA—SMARTpool, 5 nmol (L-003279-00-0005, Horizon Discovery) was used to knock down *MDM2* mRNA. Knock-down of *XPO1* mRNA and protein was verified after 24 h by RT-qPCR and immunoblot, respectively. Knock-down of *ABCB1* and *HMOX1* mRNA was verified after 24 h by RT-qPCR. Knock-down of *NEDD4L* and *MDM2* mRNA and protein was verified after 24 h by RT-qPCR and immunoblot.

**RT-qPCR.** of host cell mRNA was performed using Nucleospin RNA purification kit (Machery Nagel), on-column removal of DNA with rDNase (Machery Nagel), and the PrimeScript cDNA synthesis kit (TaKaRa, Shiga, Japan) with 400 ng RNA input in a 10 µL reaction. Sequences of PCR primers are shown in Supplementary Table S2. A LightCycler® 2.0 instrument (Roche, Mannheim, Germany) was used, employing 45 thermocycles of 95 °C for 15 s, 60 °C for 15 s, and 72 °C for 15 s. Melting curve analysis was performed to exclude artifacts resulting from primer dimer formation, using the sequence 95 °C for 15 s, 60 °C for 15 s, 95 °C for 1 min. and 37 °C for 30 s. Relative expression of target mRNAs was calculated using the $2^{-\Delta\Delta CT}$ method, using *HPRT* mRNA as internal reference.

**Assessment of *XPO1* mRNA dynamics in the presence of ActD and 4OI.** A549 cells were seeded in 12-well plates to a density of $2 \times 10^5$ cells/well. ActD (A9415, Sigma Aldrich) and/or 4OI were added to the medium to concentrations of 5 µg/ml and 100 µM, respectively. As indicated in the figure legend, ActD-containing medium was replaced after 24 h with fresh medium containing neither compound nor 100 µM 4OI. *XPO1* mRNA levels were measured by RT-qPCR after 48 h.

**Immunoblotting.** was performed as described in detail in ref. [62], using a semi-dry transfer system (Trans-Blot Turbo, BioRad), Amersham enhanced chemiluminescence western blot detection reagent (GE Healthcare Science, Pittsburgh, PA), and a Vilber fusion FX7 device (Vilber Smart Imaging, Collégien, France). The following primary antibodies were used: ACE2 (Cell Signaling Technology, 92485S, 1:1000), XPO1 (Cell Signaling Technology, 46249S, 1:1000), TMPRSS2 (Santa Cruz, sc-515727, 1:500), MDM2 (Cell Signaling Technology, 86934S, 1:1000), NEDD4L (ThermoFisher Scientific, 67276-1-IG-20UL, 1:10,000), P-STAT3 (Cell Signaling Technology, 9145S, 1:1000), STAT3 (Cell Signaling Technology, 4904S, 1:1000), and β-actin (Abcam, ab49900, 1:20,000). Goat anti-rabbit IgG-HRP (SouthernBiotech, catalogue no. 4030-05) and Goat Anti-Mouse IgG1, Human ads-HRP (SouthernBiotech, catalogue no. 1070-05) were used as secondary antibodies. ImageJ software was used to quantify the blots.

**MTT assay.** Calu-3 cells were plated in 96-well plates at a density of $2 \times 10^4$ cells per well. The MTT stock solution (Life Technologies, Cat. M6494; 5 mg ml⁻¹ in PBS) was diluted 1:10 in RPMI medium, and 50 µl of this dilution was added to each well following removal of the culture supernatant. After a 60-min incubation at 37 °C, the MTT reagent was discarded and 50 µl of dimethyl sulfoxide (DMSO; Merck) was added to solubilize the formazan crystals. Absorbance was measured using a Bio-Tek Synergy 2 plate reader at 570 nm with a 630 nm reference wavelength.

**Mitochondrial ROS assay.** 229E-luc infection (MOI = 0.3) and treatments were carried out as described above. Upon conclusion of the experiment, the cells were incubated for 5 min with medium containing 5 µM MitoSOX Red mitochondrial superoxide indicator (Invitrogen, cat# M36008). After washing with PBS, cells were resuspended with cold PBS for measurement of mitochondrial ROS by flow cytometry (Sony SP6800 ZE Analyzer, phycoerythrin channel).

**RNA sequencing.** Quality and integrity of total RNA was controlled on Agilent Technologies 2100 Bioanalyzer (Agilent Technologies; Waldbronn, Germany), using only samples with RNA Integrity Number (RIN) of ≥8 for subsequent RNA sequencing. Libraries were prepared from 500 ng total RNA using Dynabeads® mRNA DIRECT™ Micro Purification Kit (Thermo Fisher) for mRNA purification, followed by the NEBNext® Ultra™ II Directional RNA Library Prep Kit (New England BioLabs) according to the manufacturer´s protocols. The libraries were sequenced on an Illumina NovaSeq 6000 using the NovaSeq 6000 S1 Reagent Kit (100 cycles, paired-end run) yielding an average of $5 \times 10^7$ reads per RNA sample. Quality reports for each FASTQ file were generated using the FASTQC tool. Before alignment to the reference genome, raw sequences were trimmed based on base call quality and adapter contamination using fastq-mcf. Reads shorter than 15 bp were removed from the FASTQ file. Trimmed reads were aligned to the reference genome (hg38) using the open source aligner STAR[63] with settings recorded in the log file. Raw counts were generated with the R package **RSubread**[64], and features were annotated using **biomaRt** (https://www.ensembl.org/info/data/biomart/index.html). Features categorized as "rRNA" or "pseudogene" were removed from the dataset. Only features with more than 3 counts per million (CPM) in at least one set of replicates were included. Data normalization was performed using the TMM (Trimmed Mean of M-values) method[65], and normalized expression values (CPM) were log2-transformed before differential expression analysis.

**Bioinformatics.** RNAseq data were analyzed using DESeq2 (differential expression analysis, Venn diagrams, hierarchical clustering analysis and KEGG pathway GSEA). Metaboanalyst (https://www.metaboanalyst.ca/)[66] was used to prepare heatmaps. The strip charts (Fig. 8, Supplementary Fig. S7) were generated using R package ggplot2. The GSEA of XPO1 cargoes was performed with data from[20], using Webgestalt (https://www.webgestalt.org/). Network analysis of predicted E3 ligase targets was performed using UbiBrowser (http://ubibrowser.bio-it.cn/ubibrowser/home/index).

**Statistics and Reproducibility.** Data were analyzed using GraphPad Prism v8.02 (GraphPad Software), which was also used to generated most graphs. Unless stated otherwise, statistical significance was determined by ANOVA with Tukey's post-hoc test to correct for multiple hypothesis testing. Experiments were performed using three biological replicates unless stated otherwise in the figure legends. Data are shown as means +/- standard deviation (SD).

## Reporting summary
Further information on research design is available in the Nature Portfolio Reporting Summary linked to this article.

## Data availability
Bulk RNA-seq data were deposited into the GEO submission database under accession number GSE280154. The data that support the findings of this study are available from the corresponding author upon reasonable request. Uncropped western blots are shown in Supplementary Fig. S8, and numerical source data for graphs in the manuscript can be found in Supplementary Data 1–4.

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

## Acknowledgements

We thank Lothar Jänsch (Helmholtz Centre for Infection Research, HZI) for helpful discussion and Michael Jarek and the staff of the HZI Genome Analytics group for expert RNA sequencing. The study was supported by German Federal Ministry for Science and Education (BMBF) award "COVID-Protect" (01KI20143 C; to F.P., R.O., and G.G.). F.H.W. and F.Z. received salary support from BMBF award "COVID-Protect" (01KI20143C). The funders had no role in study design, data collection and analysis, decision to publish, or preparation of the manuscript. G.G. was additionally supported by the Deutsche Forschungsgemeinschaft (DFG, German Research Foundation) - Projektnummer 158989968. D.O. was supported by the

Lundbeckfonden (R335-2019-2138), the Novonordiskfonden (NNF22OC0079512), and the Danmarks Frie Forskningsfond (1026-00003B). M.C.-T. received salary from a Kræftens Bekæmpelse postdoctoral fellowship (R306-A18092).

## Author contributions

F.H.W.: methodology, investigation, formal analysis, data curation, and writing-original draft. L.C., F.Z., M.C., L.L., D.H., R.M., and J.D.: investigation and data curation. R.G.: conducted RNA sequencing. R.O., G.G., and D.O.: writing-review and editing, funding acquisition, and supervision. F.P.: writing-review and editing, supervision, funding acquisition, formal analysis, and conceptualization.

## Funding

## Competing interests

The authors declare no competing interests.

## Additional information

[1]Research Group Biomarkers for Infectious Diseases, TWINCORE Centre for Experimental and Clinical Infection Research, Hannover, Germany. [2]Research Group Biomarkers for Infectious Diseases, Helmholtz Centre for Infection Research, Braunschweig, Germany. [3]Department of Biomedicine, Aarhus University, Aarhus, Denmark. [4]Institute of Biochemistry & Research Center of Emerging Infections and Zoonoses (RIZ), University of Veterinary Medicine Hannover, Hannover, Germany. [5]Biotech Research & Innovation Centre (BRIC), University of Copenhagen, Copenhagen, Denmark. [6]Leibniz Research Laboratories for Biotechnology and Artificial Organs (LEBAO), Department of Cardiothoracic, Transplantation and Vascular Surgery (HTTG), REBIRTH-Research Center for Translational and Regenerative Medicine, Hannover Medical School, Hannover, Germany. [7]Biomedical Research in Endstage and Obstructive Lung Disease Hannover (BREATH), German Center for Lung Research (DZL), Hannover, Germany. [8]Genome Analytics, Helmholtz Centre for Infection Research, Braunschweig, Germany. [9]Centre for Individualised Infection Medicine, Hannover, Germany. [10]Present address: Gladstone Infectious Disease Institute, Gladstone Institutes, San Francisco, USA. [11]Present address: Department of One Health Virology, Wageningen Bioveterinary Research, Wageningen University & Research, Lelystad, The Netherlands. [12]Present address: Institute of Virology, Department of Hygiene, Microbiology and Public Health, Medical University of Innsbruck, Innsbruck, Austria. ✉e-mail: frank.pessler@helmholtz-hzi.de

