## [Transparent Peer Review file · Communications Biology]

NRF2 activators and the inhibitor of nuclear export, selinexor, restrict coronaviruses by targeting a network involving ACE2, TMPRSS2, and XPO1 through an NRF2-independent mechanism

Corresponding Author: Dr Frank Pessler

Version 0:

Reviewer comments:

Reviewer #1

(Remarks to the Author)

This study systematically investigated the inhibitory effects of NRF2 activators (4OI, BARD, SFN) and XPO1 inhibitors (SEL) on highly pathogenic and seasonal coronaviruses, revealing their antiviral mechanisms by downregulating the expression of ACE2, TMPRSS2, and XPO1, and clarifying that they are independent of the NRF2 signaling pathway. However, there are still key gaps that should be addressed to fully accept the model presented.

Major points:

1. Previous research (Olagnier et al. PMID: 33009401) mentioned that demonstrated that 4-octyl itaconate (4-OI) broadly inhibits viral infections (e.g., HSV1/2) in a partially NRF2-dependent manner. Please added HSV1/2 as a comparator virus to further clarify the Nrf2-independent anti-SARS-CoV-2 function of 4-OI, or discuss the possible reason for the differences.
2. The author claimed that Nrf2 agonists and SEL promotes the ACE2 degradation via ubiquitin E3 ligase NEDD4L and MDM2. Additional experiments should be conducted using proteasome inhibitors (MG132, CFZ) and lysosome inhibitors (CQ, BafA1) to confirm whether ACE2/TMPRSS2 degradation occurs via the ubiquitin-proteasome or autophagy-lysosome pathway. How about the ubiquitination level of ACE2 in different compounds and NEDD4L/MDM2-KD? Why NEDD4L knock-down attenuates the ACE2-destroying capacity of 4OI at the mRNA level? Also, does TMPRSS2 degradation due to the ubiquitination by NEDD4L and MDM2 too?
3. Since the author mentioned that SEL enhances nuclear localization of ACE2, does this enhancement contributed by the reduced nuclear export function of XPO1? Whether the inhibition of SARS-CoV-2 entry by Nrf2 agonists and SEL via ACE2/TMPRSS2 is actually due to the XPO1 downregulation? If not, it is very confusing to me to mentioned SEL function on both ACE2 and XPO1 at the same paragraph.
4. Also, how the components-blocked XPO1 hindered 229E infectivity which does not rely on ACE2/TMPRSS2?
5. Please repeat the experiments to confirm the components also inhibit 229E infectivity by Nrf2-independent manner in A549 cells.

Minor points:

1. In figure 2H-J, the immunoblot between WT/NEDD4L-KD (MDM2-KD) cells are conducted in different gels, which make readers hard to compare the ACE2 protein level between WT cells and NEDD4L-KD or MDM2-KD cells. To enhance the robustness and comparability of the data, please repeat the experiment for WT/NEDD4L-KD (figure 2H) as well as WT/MDM2-KD (Figure 2J) conditions on the same gel, and provide full-length, uncropped western blot images that include molecular weight markers and all lanes. Additionally, please ensure all the results are repeated with at least three biological replicates.
2. As previous researches mentioned (PMID: 37624697, PMID: 36862550, PMID: 30158636), 4-OI can also inhibit the activation of STING in different manner. Since SARS-CoV-2 also induced type I IFN by cGAS-STING pathway, the 4-OI-downregulated type I IFN signaling effect may due to the blockage of STING. Please discuss this possibility at the manuscript.
3. To improve clarity and avoid confusion between experimental conditions, please label Figure 5C as wildtype A549 cells and Figure 5D as ABCB1 knock-down A549 cells within the figure panels.

4. Line 452, please complete this sentence by author's name of the reference instead of directly "corroborated by [12]".

Reviewer #2

(Remarks to the Author)

Waqas and colleagues investigated the impact of the NRF2 96 activators BARD, SFN, 4OI on coronavirus infection. They report that these compounds inhibit SARS-CoV-2 infection to some degree and that 4OI also blocks expression of exportin 1 (XPO1). Similarly, an XPO1 inhibitor, SEL, is shown to exert antiviral activity. Furthermore, it is demonstrated that the compounds reduce ACE2, TMPRSS2 and XPO1 expression, that MDM2 and NEDD4L are required for 4OI- and SEL-dependent reduction in ACE2, TMPRSS2 and XPO1 expression and that 4OI and SEL reduce SARS-1 and SARS2 spike but not VSV-G-dependent cell entry. In addition, it is demonstrated that, as documented in figure 1 for SARS-CoV-2, knockout of NRF2 increases 229E infection and that the compounds reduce infection in an NRF2-independent, potentially XPO-1 dependent manner. An impact of NRF2 on ROS expression is shown but its relevance to the antiviral activity of the compounds tested was not elucidated. Further, evidence is provided that XPO1 is needed for full antiviral activity of the compounds tested, with relative XPO1-dependence varying between compounds. Finally, data supportive of 4OI reducing XPO-1 transcription are presented and the impact of 229E infection on cellular gene expression is shown, revealing that the compounds tested target similar functional pathways.

Major

This reviewer appreciates that a detailed analysis has been conducted. However, it remains largely unclear how XPO1 promotes coronavirus infection and whether the compounds tested inhibit infection mainly by targeting entry or also other steps in viral replication. The latter question should be addressed by time-of-addition assays, which will reveal if adding compounds after viruses absorb to cells still results in marked antiviral activity.

Minor

Figure 1: It is unclear why y-axes of graphs showing SARS-CoV-2 viral load were labeled in three different ways: COV2/TBP, SARS-CoV-2 viral copies and SARS-CoV-2 RNA levels. All y-axes should show the same and should be labeled the same. Please revise and state in the text whether SARS-CoV-2 mRNA or genomic RNA was measured.

Figure 1: It is essential to show that the NRF2 knock out in panel E worked.

Figure 1F: It should be stated that knock-down of NRF2 reduced spike levels and partially rescued the negative effect of 4OI on spike expression levels.

The figure legends should contain more information. In particular, it should be stated for every subpanel whether the results of a single representative experiment or the average of several experiments are shown. If a single experiment is shown, how many confirmatory experiments were conducted? If the average is shown, how many experiments were averaged? Error bars indicate SD or SEM etc?

Reviewer #3

(Remarks to the Author)

The authors investigated the role of inhibitors and NRF2 in inhibiting coronavirus replication using cell lines. They showed that NRF2 pathway and the inhibitors suppressed SARS-CoV-2 and 229E replication. The authors showed that the compounds reduced ACE2 and TMPRSS2 mRNA and protein levels. The manuscript presents a wealth of data. However, it is very difficult to follow as there are multiple pathways involved. It is unclear to me what is the major findings of the manuscript. Some of the statements are contradictory.

1. All the assay measured viral RNA. No infectious virus was measured. Viral RNA may not always correlate with virus titers.
2. The figure measuring viral RNA should be plotted in log scale.
3. In line 176, the authors showed reduction in ACE2 mRNA. However, they are claiming that ACE2 is proteasomally degraded. If there is less ACE2 mRNA then there would most likely be less ACE2 protein. So the reduction in ACE2 protein levels are due to reduced mRNA levels but not proteasomal degradation.
4. Line 168, the authors mentioned that proteasomal inhibitor MG132 led to even more rapid loss of ACE2. This suggests that the protein was not degraded through proteasomal degradation but went on to test if ACE2 was degraded by proteasomal degradation. I do not understand the rationale of these experiments.
5. The authors mentioned that XPO1 impact viral RNP export for influenza virus but also suppress CoV replication. However, CoV replication does not happen in the nucleus. Is XPO1 inhibiting CoV with a different mechanisms?
6. Does HMOX1 signal downstream of NRF2? If so the inhibition of 229E replication by HMOX1 is NRF2 dependent?

Version 1:

Reviewer comments:

Reviewer #1

(Remarks to the Author)

This study challenges the conventional understanding of NRF2 activators by revealing their potent, NRF2-independent anti-coronaviral activity through a novel host network. The revision has strengthened key claims, including the NRF2-independent function of NRF2 activators on SARS-CoV-2, the degradation manner of ACE2, and discussed the possible mechanism of XPO1 in restricting hCoV-229E. The findings are of significant interest to the fields of virology and cell signaling.

To elevate the manuscript to the highest level of mechanistic clarity and the expected quality for Communications Biology, I consider one adjustment may be helped to tighten the logic:

The current framing, while engaging, creates a slight disconnect between the title "NRF2 activators" and the central NRF2-independent conclusion. This could lead to conceptual confusion. To align the title with the core message, I recommend refining the title. One effective solution would be to add a clarifying phrase such as '...through an NRF2-independent mechanism'.in title. Alternatively, the authors and editors may consider other formulations that capture this pivotal paradox.

Reviewer #2

(Remarks to the Author)

The authors have largely addressed the reviewer's comments. However, Figures 2D–G should be subjected to appropriate statistical analysis, and labels such as "Original Freiburg DMSO" should be replaced with scientifically precise terminology.

Reviewer 1	
This study systematically investigated the inhibitory effects of NRF2 activators (4OI, BARD, SFN) and XPO1 inhibitors (SEL) on highly pathogenic and seasonal coronaviruses, revealing their antiviral mechanisms by downregulating the expression of ACE2, TMPRSS2, and XPO1, and clarifying that they are independent of the NRF2 signaling pathway. However, there are still key gaps that should be addressed to fully accept the model presented.	
1. Previous research (Olagnier et al. PMID: 33009401) mentioned that demonstrated that 4-octyl itaconate (4-OI) broadly inhibits viral infections (e.g., HSV1/2) in a partially NRF2-dependent manner. Please add HSV1/2 as a comparator virus to further clarify the Nrf2-independent anti-SARS-CoV-2 function of 4-OI, or discuss the possible reason for the differences.	We thank the reviewer for this insightful comment. To address the concern regarding the NRF2-independent antiviral activity of 4-OI against SARS-CoV-2, we have performed additional experiments in WT and NRF2-KO 786-O cells and consistently observed that the antiviral effect of 4-OI against SARS-CoV-2 is maintained in the KO cells. These data strengthen the conclusion that 4-OI exerts its anti-SARS-CoV-2 activity to a substantial extent independently of NRF2. Results, lines 147-151. Figure 2H. Discussion, lines 548-554. Regarding the reviewer's request to include HSV-1/2 as comparator viruses, prior work (Olagnier et al., PMID: 33009401) indeed showed that 4-OI displays partially NRF2-dependent antiviral activity against HSV-1/2. We agree that this reflects a mechanistic difference between SARS-CoV-2 and herpesviruses. By contrast, our new NRF2-deficiency experiments demonstrate that SARS-CoV-2 replication is inhibited by 4-OI through mechanisms that do not require NRF2, potentially reflecting differential viral protein sensitivities or pathways engaged by electrophilic stress. We would prefer to not add a comparative experiment with HSV1/2 to the manuscript, as it is already very long and the suggested experiments would need to be performed through a collaboration with an HSV lab, which would further delay completion of the manuscript. We added text to Discussion regarding the possible differences between HSV and CoV with respect to 4OI function, i.e. the greater NRF2 dependence for HSV. Discussion, lines 505-509.
2. The author claimed that Nrf2 agonists and SEL promotes the ACE2 degradation via ubiquitin E3 ligase NEDD4L and MDM2. Additional experiments should be conducted using proteasome inhibitors (MG132, CFZ) and lysosome inhibitors (CQ, BafA1) to confirm whether ACE2/TMPRSS2 degradation occurs via the ubiquitin-proteasome or autophagy-lysosome pathway.	We thank the reviewer for this very helpful suggestion. The surprising finding that MG132 exacerbated ACE2 loss was in the original version of the manuscript. We have additionally tested CFZ. Due to toxicity, we could only look at the 6 h time point with CFZ, but it is apparent that (like MG132) it destabilizes ACE2 long before 4-OI and SEL. The likely explanation is that inhibiting the proteasome induces degradation by the lysosomal pathway. We tested the lysosomal inhibitors CQ and BafA1, and both reduced the ACE2-destabilizing activity of 4OI and SEL. The results strongly suggest that the lysosomal pathway plays a major role in 4-OI mediated loss of ACE2. Results, lines 198-207.

	Figure 3J-N. Discussion, lines 463-472.
How about the ubiquitination level of ACE2 in different compounds and NEDD4L/MDM2-KD?	We regret that we could not assess the ubiquitination level of ACE2 due to limitations of resources and length of the manuscript. However, the circumstantial evidence that the effect is mediated by ubiquitination is strong. Nonetheless, we added to Discussion that we would have to determine ubiquitination levels of ACE2 if we wanted to infer that ACE2 is a direct target of the studied E3 ligases. Discussion, lines 472-477.
Why NEDD4L knock-down attenuates the ACE2-destroying capacity of 4OI at the mRNA level?	Our hypothesis was that part of the mRNA reduction by 4OI is mediated by affecting a transcription factor on the ACE2 promoter. STAT3 is a major transcription factor on the ACE2 promoter and it is at least in part regulated by ubiquitination. We tested whether 4-OI diminishes levels of STAT3 and/or p-STAT3 (the active form). Only levels of p-STAT3 went down, which argues against the ubiquitination hypothesis. Clearly, additional work is required to elucidate the mechanism. We expanded the relevant text about this in Discussion. Results, lines 219-223. Figure 3R. Discussion, lines 480-484.
Also, is TMPRSS2 degradation due to the ubiquitination by NEDD4L and MDM2 too?	We tried, but decided to abort the experiments because of poor quality of the antibody. Since the focus of our paper is on ACE2, and to not increase the length of the manuscript further, we decided to not pursue this question further at this time. We added to Discussion that this should be investigated in the future (as well as any impact on XPO1 ubiquitination by 4OI and SEL). Discussion, lines 474-475.
4. Since the author mentioned that SEL enhances nuclear localization of ACE2, does this enhancement contributed by the reduced nuclear export function of XPO1? Whether the inhibition of SARS-CoV-2 entry by Nrf2 agonists and SEL via ACE2/TMPRSS2 is actually due to the XPO1 downregulation? If not, it is very confusing to me to mentioned SEL function on both ACE2 and XPO1 at the same paragraph.	Our results clearly show that the major effect of 4-OI is due to lysosomal destruction of ACE2. Since 4-OI and SEL had such similar effects on ACE2 and TMPRSS2 protein levels and since they both recognize XPO1 as a major binding site, we do believe that the effect is in some way mediated by XPO1 inhibition. However, at this point we cannot pinpoint the exact mechanism. We improved the corresponding text in Discussion, as well as the Graphical Abstract. Discussion, lines 455-470. Graphical Abstract
5. Also, how the components-blocked XPO1 hindered 229E infectivity which does not rely on ACE2/TMPRSS2?	We were not able to identify a molecular mechanism for how the compounds inhibit 229E via XPO1. We expanded the 229E-related text in Discussion, particularly adding that a direct antiviral effect of XPO1 blockage (via targeting nuclear cytoplasmic shuttling of viral proteins, as has been suggested for SARS-CoV-1/2) is not supported by current data.

	Discussion, lines 510-519.
5. Please repeat the experiments to confirm the components also inhibit 229E infectivity by Nrf2-independent manner in A549 cells.	We performed the suggested experiment using NRF2 KO A549 cells. The results agreed well with our previous results in NRF2 KO ECs in that the antiviral effect was largely NRF2 independent. Results, lines 307-309. Figure 5C,D.
Minor	
1. In figure 2H-J, the immunoblot between WT/NEDD4L-KD (MDM2-KD) cells are conducted in different gels, which make readers hard to compare the ACE2 protein level between WT cells and NEDD4L-KD or MEM2-KD cells. To enhance the robustness and comparability of the data, please repeat the experiment for WT/NEDD4L-KD (figure 2H) as well as WT/MDM2-KD (Figure 2J) conditions on the same gel, and provide full-length, uncropped western blot images that include molecular weight markers and all lanes. Additionally, please ensure all the results are repeated with at least three biological replicates.	We regret that due the limited resources and the extensive additional experiments required for other aspects of the paper, we could not repeat this experiment, as it would require new knock-downs of both targets and performing the experiments 3 independent times. In our opinion, even from the blots in their current state it is evident that the ACE2 protein levels, under 4OI and SEL treatment, are substantially higher in the KD cells. Considering the lack of replicates, we toned down the text, and replaced “nearly abrogated” with “strongly reduced”. Figure 3H,I.
2. As previous researches mentioned (PMID: 37624697, PMID: 36862550, PMID: 30158636), 4-OI can also inhibit the activation of STING in different manner. Since SARS-CoV-2 also induced type I IFN by cGAS-STING pathway, the 4-OI-downregulated type I IFN signaling effect may due to the blockage of STING. Please discuss this possibility at the manuscript.	We added these references to the respective text in Results, lines 247-249. Discussion, lines 486-490.
3. To improve clarity and avoid confusion between experimental conditions, please label Figure 5C as wildtype A549 cells and Figure 5D as ABCB1 knock-down A549 cells within the figure panels.	We corrected the figure labels. Figure 4C,D.
4. Line 452, please complete this sentence by author’s name of the reference instead of directly “corroborated by [12]”.	Correct as suggested Discussion, line 443.
Reviewer 2	
Waqas and colleagues investigated the impact of the NRF2 96 activators BARD, SFN, 4OI on coronavirus infection. They report that these compounds inhibit SARS-CoV-2 infection to some degree and that 4OI also blocks expression of exportin 1 (XPO1). Similarly, an XPO1 inhibitor, SEL, is shown to exert antiviral activity. Furthermore, it is demonstrated that the compounds reduce ACE2, TMPRSS2 and XPO1 expression, that MDM2 and NEDD4L are required for 4OI- and SEL-dependent reduction in ACE2, TMPRSS2 and XPO1 expression and that 4OI and SEL reduce SARS-1 and SARS2 spike but not VSV-G-dependent cell entry. In addition, it is demonstrated that, as documented in figure 1 for SARS-CoV-2, knockout of NFR2 increases 229E infection and that the compounds reduce infection in an NRF2-independent, potentially XPO-1 dependent manner. An impact of NRF2 on ROS expression is shown but its relevance to the antiviral activity of the compounds tested was not elucidated. Further, evidence is provided that XPO1 is needed for full	

antiviral activity of the compounds tested, with relative XPO1-dependence varying between compounds. Finally, data supportive of 4OI reducing XPO-1 transcription are presented and the impact of 229E infection on cellular gene expression is shown, revealing that the compounds tested target similar functional pathways.	
This reviewer appreciates that a detailed analysis has been conducted. However, it remains largely unclear how XPO1 promotes coronavirus infection and whether the compounds tested inhibit infection mainly by targeting entry or also other steps in viral replication. The latter question should be addressed by time-of-addition assays, which will reveal if adding compounds after viruses absorb to cells still results in marked antiviral activity.	We performed the suggested time-of-addition experiment. The results showed that the strongest antiviral effect of 4OI and SEL is seen before infection, but that there is an additional (weaker) effect post-infection. All experiments were done 2-3 times Results, lines 143-151. Figure 2A-H.
Minor	
Figure 1: It is unclear why y-axes of graphs showing SARS-CoV-2 viral load were labeled in three different ways: COV2/TBP, SARS-CoV-2 viral copies and SARS-CoV-2 RNA levels. All y-axes should show the same and should be labeled the same. Please revise and state in the text whether SARS-CoV-2 mRNA or genomic RNA was measured.	We harmonized the x-axis labels and added the information that it measures genomic RNA with a contribution of subgenomic RNA as well as the reference to the RNA calculation method. Figure 1 legend Methods, lines 635-640.
Figure 1: It is essential to show that the NRF2 knock out in panel E worked.	It is notoriously difficult to obtain a decent NRF2 immunoblot from Calu-3 cells, and we ran into the same problem. We, therefore, deleted panels E and F from the revised figure and are showing NRF2 independence of 4OI in Figure 2, using previously published 786-O cells with a stable CRISPR NRF2-KO. Figure 2H.
Figure 1F: It should be stated that knock-down of NRF2 reduced spike levels and partially rescued the negative effect of 4OI on spike expression levels. The figure legends should contain more information. In particular, it should be stated for every subpanel whether the results of a single representative experiment or the average of several experiments are shown. If a single experiment is shown, how many confirmatory experiments were conducted? If the average is shown, how many experiments were averaged? Error bars indicate SD or SEM etc?	We updated the figure legends accordingly.
Reviewer 3	

The authors investigated the role of inhibitors and NRF2 in inhibiting coronavirus replication using cell lines. They showed that NRF2 pathway and the inhibitors suppressed SARS-CoV-2 and 229E replication. The authors showed that the compounds reduced ACE2 and TMPRSS2 mRNA and protein levels. The manuscript presents a wealth of data. However, it is very difficult to follow as there are multiple pathways involved. It is unclear to me what is the major findings of the manuscript. Some of the statements are contradictory.	
1. All the assay measured viral RNA. No infectious virus was measured. Viral RNA may not always correlate with virus titers.	In the order-of-addition experiment we have also assessed TCID50 following 4-OI and Selnexor treatment. Figure 2D-G
2. The figure measuring viral RNA should be plotted in log scale.	We transformed Figure 1F to log10 scale. However, we prefer to leave panel G on linear scale for clarity sake.
3. In line 176, the authors showed reduction in ACE2 mRNA. However, they are claiming that ACE2 is proteasomally degraded. If there is less ACE2 mRNA then there would most likely be less ACE2 protein. So the reduction in ACE2 protein levels are due to reduced mRNA levels but not proteasomal degradation.	The cycloheximide chase experiment clearly showed that the major effect of 4-OI is post-translational. Considering that 4-OI shortens the half-life of ACE2 from about 12 h to much less than 3 h (Figure 3E-G), reducing mRNA (even if it were immediate) could not account for the rapid disappearance of ACE2. A presumed silencing of the promoter would, therefore, not be solely responsible for the rapid turnover of ACE2 but would likely contribute to this effect. This is explained better now in Discussion. Discussion, lines 457-477.
4. Line 168, the authors mentioned that proteasomal inhibitor MG132 led to even more rapid loss of ACE2. This suggests that the protein was not degraded through proteasomal degradation but went on to test if ACE2 was degraded by proteasomal degradation. I do not understand the rationale of these experiments.	The finding that MG132 actually sped up ACE2 loss makes more sense now that we have data suggesting that ACE2 is destroyed by the lysosomal pathway (which can be activated by MG132). Figure 3K-N. Results, lines 203-207. Discussion, lines 467-477.
5. The authors mentioned that XPO1 impact viral RNP export for influenza virus but also suppress CoV replication. However, CoV replication does not happen in the nucleus. Is XPO1 inhibiting CoV with a different mechanisms?	Our data indicate that XPO1 blockade interferes with SARS-CoV-2 infection by a very similar mechanism as 4-OI, i.e. primarily by interfering with viral entry by downregulating ACE2. The antiviral MoA of Selnexor against CoV229E is largely unclear. We have expanded on this in Discussion Discussion, lines 512-519
6. Does HMOX1 signal downstream of NRF2? If so the inhibition of 229E replication by HMOX1 is NRF2 dependent?	HMOX1 expression is largely NRF2 dependent in the EC model used. But since the antiviral effects of the compounds are NRF2 independent, its contribution to the antiviral effects must be quite small. We changed the text in Results so this is expressed more clearly. Results, lines 330-340. Discussion, lines 550-560.